# In vivo bioluminescence imaging of natural bacteria within deep tissues *via* ATP-binding cassette sugar transporter

Qian Zhang[1,3], Bin Song[1,3], Yanan Xu[1,3], Yunmin Yang[1], Jian Ji[2], Wenjun Cao[2], Jianping Lu[1], Jiali Ding[1], Haiting Cao[1], Binbin Chu[1], Jiaxu Hong[2] ✉, Houyu Wang ®[1] ✉ & Yao He ®[1] ✉

Most existing bioluminescence imaging methods can only visualize the location of engineered bacteria in vivo, generally precluding the imaging of natural bacteria. Herein, we leverage bacteria-specific ATP-binding cassette sugar transporters to internalize luciferase and luciferin by hitchhiking them on the unique carbon source of bacteria. Typically, the synthesized bioluminescent probes are made of glucose polymer (GP), luciferase, Cy5 and ICG-modified silicon nanoparticles and their substrates are made of GP and D-luciferin-modified silicon nanoparticles. Compared with bacteria with mutations in transporters, which hardly internalize the probes in vitro (i.e., ~2% of uptake rate), various bacteria could robustly engulf the probes with a high uptake rate of around 50%. Notably, the developed strategy enables ex vivo bioluminescence imaging of human vitreous containing ten species of pathogens collected from patients with bacterial endophthalmitis. By using this platform, we further differentiate bacterial and non-bacterial nephritis and colitis in mice, while their chemiluminescent counterparts are unable to distinguish them.

The microorganisms have many vital roles in health and disease[1-6]. Bacterial activity in vivo is heavily influenced by their location within the host organism. The existing clinical imaging methods such as computed tomography (CT), magnetic resonance imaging (MRI) and can provide non-invasive imaging of bacterial infections in the body. However, due to their relatively poor selectivity, they are unable to distinguish inflammation caused by bacterial infections from inflammation caused by other causes such as cancer or autoimmune diseases[7-10]. Recently, optical imaging techniques can provide localized, qualitative and quantitative bacterial information at the molecular level[11-14]. As the most widely used optical imaging method, fluorescence imaging requires real-time optical excitation, however, it may lead to background autofluorescence of biological tissues, resulting in a relatively poor signal-background ratio[15-17].

Ascribed to the elimination of an external light irradiation, bioluminescence imaging (BLI) features several advantages over fluorescence imaging, such as lower background, higher sensitivity[18-28]. To date, the bioluminescent systems for bacteria detection can be basically categorized into two types: (1) the endogenous BLI system, which involves the genetic engineering of bacteria to express luciferases[29-33]; (2) the exogenous BLI system, in which the self-luminescence comes from the oxidation of the exogenous luciferin or caged-luciferin substrates catalysed by the exogenous luciferase in the presence of bacterial adenosine triphosphate (ATP) produced by the lysis of bacteria[34]. In spite of the great progress of BLI, it has intrinsic shortcomings in the bacterial imaging, as follow: "first, the endogenous BLI systems need genetic modification of bacteria; second, the exogenous BLI systems need

[1]Suzhou Key Laboratory of Nanotechnology and Biomedicine, Institute of Functional Nano & Soft Materials & Collaborative Innovation Center of Suzhou Nano Science and Technology (NANO-CIC), Soochow University, Suzhou 215123, China. [2]Department of Ophthalmology and Vision Science, Shanghai Eye, Ear, Nose and Throat Hospital, Fudan University, Shanghai, China. [3]These authors contributed equally: Qian Zhang, Bin Song, Yanan Xu. ✉e-mail: jiaxu.hong@fdeent.org; houyuwang@suda.edu.cn; yaohe@suda.edu.cn

the destruction of the bacterial cells to consume the intracellular ATP", thus being unable to image live bacteria[29–34].

A theoretically promising but methodologically undeveloped way to visualize various natural bacteria in vivo with bioluminescence is to selectively deliver bioluminescent reporter into bacterial cells, directly consuming the ATP inside the bacteria. Intriguingly, the "Trojan horse" antibiotic strategy could deliver siderophore-linked antibiotics into bacterial cytoplasm through the bacteria iron transporters[35–41]. In our previous work, we have demonstrated that glucose polymer (GP) modified nanoprobes can be internalized into various bacteria through bacteria-specific ABC transporter pathway, as summarized in Supplementary Table 1[42–45]. However, the use of a "Trojan horse" strategy to selectively deliver bioluminescent indicators into bacteria has not, to the best of our knowledge, been reported before.

To fill the technical gap, we set out to selectively deliver luciferase and luciferin into diverse natural bacteria through ATP-binding cassette (ABC) sugar transporters, by hitchhiking them on α (1-4)-glucosidically linked glucose polymers (dextrose equivalent of 4.0-7.0)-linked nanoparticles. Figure 1 schematically illustrates that both Gram-negative and Gram-positive bacteria could actively engulf the synthesized bioluminescent probes (i.e., glucose polymer (GP), luciferase, Cy5 and ICG-modified silicon nanoparticles (GP-Si-BPs)) and their substrates (GP and D-luciferin-modified silicon nanoparticles (GP-Si-Luc)). GP (e.g., *poly[4-O-(α-D-glucopyranosyl)-D-glucopyranose]*) serves as the ubiquitous carbon source, which can be robustly internalized into bacterial cells through the ABC sugar transporter[9,42,46]. Using the ABC sugar transporter in *Escherichia coli* (*E. coli*) as an example, it has five subunits: LamB, MalE, MalF, MalG and MalK. Among these subunits, LamB is a typical outer membrane diffusion porin, and MalE can recognize α (1-4)-glucosidically linked GP molecules[47–53]. By leveraging this uptake mechanism, small-size (e.g., ~5 nm) GP-modified nanoparticles including silicon nanoparticles, gold nanoparticles and carbon dots have recently been demonstrated to be selectively and robustly internalized into bacterial cells[43–45]. Analogously, here we show that diverse bacteria eat their camouflaged 'foods', i.e., GP-Si-BPs and GP-Si-Luc. After internalization into bacterial cells, the luciferin in GP-Si-Luc is directly activated by ATP within bacteria, followed by the oxidation catalysed by the luciferase in GP-Si-BPs. By further employing an energy transfer relay, that integrates bioluminescence resonance energy transfer (BRET) between luciferase and Cy5 and fluorescence resonance energy transfer (FRET) between Cy5 and ICG,

the developed Trojan BLI probes enable near-infrared (NIR) imaging of bacteria within deep tissues. Furthermore, the loading of ICG allows the photothermal killing of bacteria under NIR irradiation. We demonstrate the presented Trojan horse BLI strategy allows ex vivo bioluminescence imaging of ten species of bacteria in the vitreous of patients with bacterial endophthalmitis. We also demonstrate that the presented Trojan horse BLI strategy enables not only selective and sensitive imaging but also photothermal therapy of pathogens in deep tissues, in proof-of-concept models of bacterial nephritis and colitis in mice.

## Results

### Characterization of Trojan BLI probes

Supplementary Fig. 1 illustrated the synthetic route for GP-Si-BPs and GP-Si-Luc step by step. Briefly, we first synthesized GP-conjugated SiNPs (GP-SiNPs) through the Schiff base reaction, in which the Schiff base was formed based on the reaction between the aldehyde groups of GP (0.56 mM, 100 μL) and the amino groups on the surface of SiNPs (0.08 mM, 200 μL). Next, by leveraging electrostatic adsorption, the dye molecules of Cy5 (0.56 mM, 2 μL) and ICG (0.56 mM, 12 μL) were adsorbed onto GP-SiNPs. By further conjugating with firefly luciferase (0.06 mM, 100 μL) via *N*-(3-dimethylaminopropyl)-*N'*-ethyl-carbodiimide hydrochloride (EDC)/*N*-hydroxysuccinimide (NHS)–activated condensation reaction, we finally obtained the bioluminescent probes of GP-Si-BPs. Analogously, by conjugating GP-SiNPs with D-luciferin (0.30 mM, 100 μL) based on the EDC/NHS condensation reaction, we synthesized the bioluminescent substrates of GP-Si-Luc. The chemical yield of GP-Si-BPs was 76.1% and the chemical yield of GP-Si-Luc was 78.8%, which were calculated according to the reported protocol[54]. The amounts of GP, Cy5, ICG, luciferase, D-luciferin loaded onto SiNPs were strictly quantified by the corresponding calibration absorption curves (Supplementary Fig. 2). Typically, the actual amount of Cy5 adsorbed onto the GP-SiNPs was ~0.0044 mM, and the actual amount of ICG adsorbed onto the GP-SiNPs was ~0.026 mM. To obtain the equivalent dose, the detected absorbance of Cy5 and ICG should be kept the same among groups.

As revealed in the transmission electron microscopy (TEM) images and high-resolution TEM image of SiNPs, GP-Si-BPs and GP-Si-Luc (Fig. 2a) and the corresponding size distribution (Supplementary Fig. 3), the average diameter of GP-Si-BPs was ~2.8 nm and the average diameter of GP-Si-Luc was ~2.9 nm, both of which was slightly larger

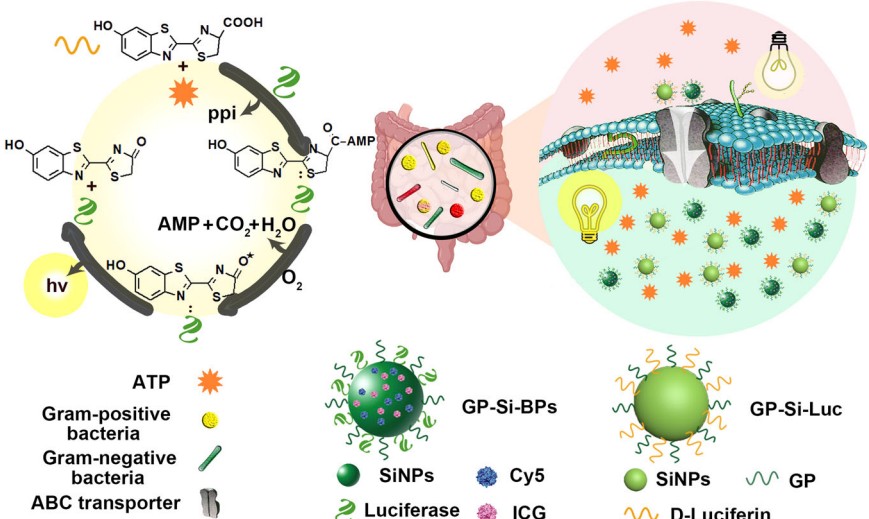

**Fig. 1 | Schematic design of ABC sugar transporter enabling selective delivery of bioluminescent nanoprobes into Gram-positive bacteria and Gram-negative bacteria to visualize various natural bacteria in vivo with bioluminescence by** directly consuming the ATP inside the bacteria. The nanoprobes are made of GP, Cy5, ICG and luciferase-modified silicon nanoparticles (SiNPs) (GP-Si-BPs) and GP, D-luciferin-modified SiNPs (GP-Si-Luc).

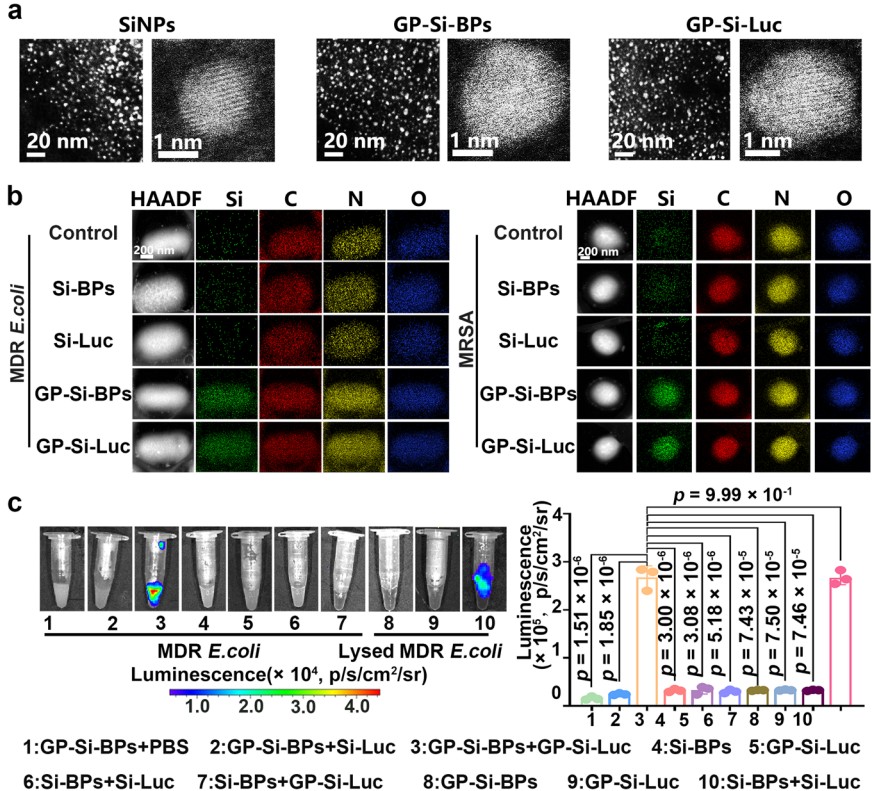

**Fig. 2 | Characterization of Trojan BLI probes. a** TEM images and high-resolution TEM images of SiNPs, GP-Si-BPs and GP-Si-Luc. Scale bar, 20 nm and 1 nm. **b** High-angle annular dark field-scanning TEM (HAADF-STEM) images of MDR *E. coli* or MRSA treated by PBS, 0.06 mM of Si-BPs, Si-Luc, GP-Si-Luc, GP-Si-BPs at 37 °C for 2.5 h. After incubation, the treated bacteria were rinsed with PBS buffer for several times. The bacterial cell concentration is ~1.0 × 10⁷ CFU. Scale bar, 200 nm. **c** Bioluminescent signals of MDR *E. coli* with different treatments as indicated. After incubation, the treated bacteria were rinsed with PBS buffer for several times, followed by imaging (IVIS Lumina III). All imaging experiments were repeated three times with similar results. Statistical analysis was performed using a one-way ANOVA analysis. Data are presented as mean values +/− SD ($n = 3$). Source data are provided as a Source data file. The cartoons are created by Prof. Houyu Wang.

than that of naked SiNPs (e.g., ~2.2 nm). The hydrodynamic diameter of GP-Si-BPs, GP-Si-Luc and pure SiNPs was ~5.6 nm, ~4.1 nm and ~3.1 nm, measured by dynamic light scattering (DLS) (Supplementary Fig. 4). The particle size measured by DLS is slightly larger than that in TEM, which may be due to different surface states of the same sample under the two measurement conditions. Specifically, the solvent in the sample must be strictly removed for TEM characterization, thus yielding a smaller diameter than that measured by DLS, as discussed in previous reports[55,56]. Additionally, we have systematically demonstrated the good stability of luciferase and luciferin after SiNPs conjugation (Supplementary Fig. 5).

To primarily interrogate whether GP-Si-BPs and GP-Si-Luc could enter natural bacteria, two clinically-derived multidrug-resistant *Escherichia coli* (MDR *E. coli*) and multidrug-resistant *Staphylococcus aureus* (MRSA) cells were isolated for the following experiments. They were obtained from patients with keratitis who were diagnosed and treated in the Shanghai Eye, Ear, Nose and Throat Hospital, Fudan University. The isolated strains (~1.0 × 10⁷ CFU) were incubated with GP-Si-BPs (0.06 mM), GP-Si-Luc (0.06 mM) at 37 °C for 2.5 h, then were washed by PBS buffer for several times. The scanning electron microscope (SEM) images in Supplementary Fig. 6 showed that the surface and morphology of the treated MDR *E. coli* and MRSA cells were comparable to untreated ones. As displayed in the elemental mapping in high-angle annular dark field-scanning transmission electron microscope (HAADF-STEM) images (Fig. 2b), carbon, nitrogen and oxygen elements appeared in each group, while silicon elements existed only in the bacteria treated with GP-Si-BPs or GP-Si-Luc. Apparently, the observed silicon signals were assigned to SiNPs in GP-Si-BPs or GP-Si-Luc, thus directly demonstrating the internalization of GP-Si-BPs or GP-Si-Luc into bacterial cells. Finally, the ex vivo BLI of

treated bacteria revealed that the distinct luminescence was only found in GP-Si-BPs+ GP-Si-Luc group, which was around ~10-fold stronger than other groups (Fig. 2c). The high brightness was due to the amount of ATP, which was necessary to activate luciferin in the luciferase-mediated light-emitting reaction, exactly about 10 times higher inside the bacteria than outside (Supplementary Fig. 7). The ATP content was determined by the ATP assay kit. Interestingly, the comparable bioluminescent signals were observed in the lysed bacteria upon addition of probes without modification of GP molecules (Si-BPs + Si-Luc) (Fig. 2c). We also treated the bacteria with the ATP inhibitor of DCC (Dicyclohexylcarbodiimide) before incubation with GP-Si-BPs + GP-Si-Luc. As expected, no distinct signals appeared (Supplementary Fig. 8). These results together proved that the constructed Trojan BLI probes indeed entered bacteria, and the brightness of the produced bioluminescence was dependent on the amount of ATP inside the bacteria.

## Evaluation of Trojan BLI probes in vitro

In the developed bioluminescent probes, the core SiNPs served as the vectors to load bioluminescent indicators. The bioluminescent indicators included the self-luminescence source of luciferase ($\lambda_{em} = 560$ nm), and two organic dyes of Cy5 ($\lambda_{ex} = 646$ nm, $\lambda_{em} = 664$ nm) and ICG ($\lambda_{ex} = 780$ nm, $\lambda_{em} = 840$ nm) (Fig. 3a). Benefited from the enough overlap in emission and absorption bands, the NIR emission can be achieved based on a dual resonance energy transfer relay process, combining BRET between luciferase and Cy5, and FRET between Cy5 and ICG (Fig. 3b). As further revealed in Fig. 3c, when the ratio of Cy5 to ICG was 1:6, the highest energy transfer efficiency can be achieved. Typically, the optimal FRET efficiency between Cy5 and ICG was determined to be 32% and the corresponding Förster

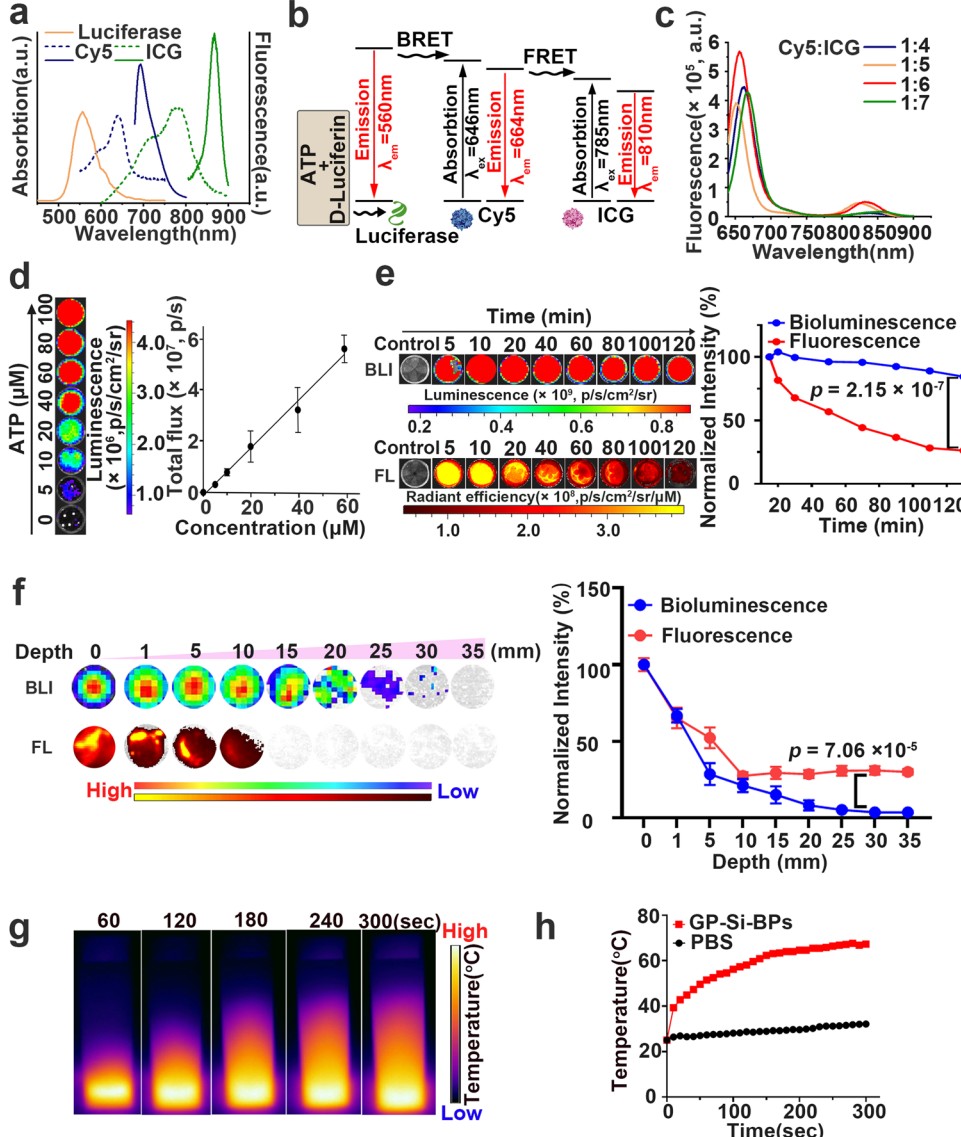

**Fig. 3 | Evaluation of Trojan BLI probes in vitro. a** UV-vis absorption and emission spectra of Cy5, ICG and bioluminescence of luciferase. Dashed and solid lines refer to absorption and emission spectra, respectively. **b** A scheme illustrating the mechanism of an energy transfer relay integrating BRET and FRET in the developed strategy. **c** Fluorescence spectra of GP-Si-BPs with different ratios of Cy5 to ICG. **d** Bioluminescent signals of GP-Si-BPs + GP-Si-Luc as a function of the ATP concentration (mean ± SD, $n = 3$). **e**, **f** Time-resolved (**e**) and chicken breast tissue thickness-dependent (**f**) bioluminescence and fluorescence (excitation: 780 nm) images and corresponding normalized signal for quantitative comparison of bioluminescence and fluorescence signal change. The normalized signal in (**e**) is defined as the ratio of the emission intensity detected at any time to the emission intensity detected at 5 min. The normalized signal in (**f**) is defined as the ratio of the emission intensity detected at any tissue thickness to the emission intensity detected at 0 mm. The bioluminescence is produced by the mixture of GP-Si-BPs (0.06 mM), D-luciferin (150 μM) and ATP (10 μM). Data are presented as mean values +/− SD ($n = 3$). **g** The photothermal heating images of PBS, GP-Si-BPs under the irradiation of 808-nm laser. **h** The photothermal heating curves of PBS, GP-Si-BPs under the irradiation of 808-nm laser. All imaging experiments were repeated three times with similar results. Statistical analysis was performed using a one-way ANOVA analysis. Source data are provided as a Source data file.

distance ($R_0$) was calculated to be 1.03 nm, ensuring the efficient emission in NIR region. As revealed in Fig. 3d, the bioluminescent intensity of GP-Si-BPs + GP-Si-Luc linearly enhanced with the increase of ATP concentration, suggesting ATP content-dependent manner of GP-Si-BPs. Upon addition of D-Luciferin (150 μM) and ATP (10 μM), the photostability of GP-Si-BPs (0.06 mM) was further tested. As revealed in Fig. 3e, the BLI system exhibited a relatively good bioluminescence stability, retaining 84% of normalized signal after 120 min because enzymatic reaction between luciferin and luciferase was stably maintained as it wasn't affected by external energy (e.g., light irradiation). On the contrary, the normalized signal of ICG in GP-Si-BPs sharply declined by 26% under continuous 808-nm irradiation for 120 min because continuous 808-nm irradiation to ICG was responsible for

potential bleaching and quenching effect for reduced emission. Afterwards, we examined the tissue penetration depth of GP-Si-BPs. Experimentally, GP-Si-BPs (0.06 mM) mixed with D-Luciferin (150 μM) and ATP (10 μM) in a black 96-well plate was covered by the chicken breast tissue with various thickness, followed by recording bioluminescent signals and fluorescent signals under 808-nm irradiation. As revealed in Fig. 3f, the normalized bioluminescent signal declined along with the increase of chicken breast tissue thickness ranging from 0 to 35 mm. Typically, when the thickness was up to 30 mm, the normalized bioluminescent intensity of GP-Si-BPs was still significantly higher than the ICG-based NIR emission imaging group ($p < 0.0001$). Analogously, the tissue penetration depth of GP-Si-BPs was significantly larger than that of GP-Ce6-SiNPs, as indicated in

Supplementary Fig. 9. Typically, when the thickness was 5 mm, the fluorescent signal of GP-Ce6-SiNPs was undetectable. The good penetration depths of GP-Si-BPs lay foundation in the subsequent in vivo imaging applications in deep tissues. Finally, we tested the photothermal ability of GP-Si-BPs in vitro by using an IR thermal imaging camera. As shown in Fig. 3g, h, the temperature of GP-Si-BPs solution (0.06 mM) could rapidly climb to 61 °C under 808-nm laser irradiation for 5 min, suggesting the potential photothermal therapy (PTT) of GP-Si-BPs against bacteria.

### Trojan BLI probes targeting diverse bacteria

To optimize the amount of linked GP and incubation time, the uptake efficiency of GP-Si-BPs by bacteria was systematically investigated by using flow cytometry analysis. As revealed in Supplementary Fig. 10, the uptake rate of ~1.0 × 10$^7$ CFU of MDR $E. coli$ reached to its maximum value (e.g., 48.5% of MRSA and 48.8% of MDR $E. coli$) when the bacteria were incubated with the GP-Si-BPs containing 0.56 mM GP concentration for 2.5 h. The uptake rate did not enhance significantly when further increasing the GP concentration and the incubation time, indicating a saturation state has been achieved under this circumstance. The similar trends were also observed in GP-Si-Luc. Accordingly, 0.56 mM GP and 2.5 h incubation were employed in the following experiments.

Next, we demonstrated whether GP-Si-BPs and GP-Si-Luc could target various natural bacteria. To this end, we selected two Gram-negative bacteria, i.e., clinically-derived MDR $E. coli$, $Salmonella typhimurium$ (STm) and two Gram-positive bacteria, i.e., clinically-derived MRSA and $Staphylococcus aureus$ ($S. aureus$). As revealed in confocal laser scanning microscopy (CLSM) images in Fig. 4a and Supplementary Fig. 11a, we could observe the distinct red fluorescent signals (assigned to ICG, $\lambda_{ex} = 780$ nm, $\lambda_{em} = 800\text{-}840$ nm) in MRSA, MDR $E. coli$, $S. aureus$ and STm after 2.5 h of incubation with GP-Si-BPs (0.06 mM). Quantitatively, the corresponding flow cytometry revealed that the uptake efficiency of GP-Si-BPs was 52.7% by MRSA, 49.6% by MDR $E. coli$, 53.5% by $S. aureus$ and 47.1% by STm, respectively. Analogously, we could detect strong green fluorescent signals (assigned to D-luciferin, first column, $\lambda_{ex} = 328$ nm, $\lambda_{em} = 530\text{-}560$ nm) in MRSA, MDR $E. coli$, $S. aureus$ and STm after 2.5 h of incubation with GP-Si-Luc (0.06 mM) (Fig. 4b and Supplementary Fig. 11b). Also, the relative high uptake efficiency of GP-Si-Luc was determined in MRSA (48.1%), MDR $E. coli$ (45.7%), $S. aureus$ (47.2%), and STm (48.9%) respectively. As expected, both green and red signals can be observed in the MRSA, MDR $E. coli$, $S. aureus$ and STm treated with GP-Si-BPs (0.06 mM) and GP-Si-Luc (0.06 mM) (GP-Si-BPs + GP-Si-Luc) for 2.5 h (Fig. 4c and Supplementary Fig. 11c). On the contrary, we can't observe detectable fluorescent signals in the bacteria treated with the nanoagents without the modification of GP molecules (i.e., Si-Luc, Si-BPs or Si-Luc + Si-BPs) (Supplementary Figs. 12 & 13). These results primarily confirmed the bacteria-targeting ability of GP ligands.

To further demonstrate whether GP-Si-BPs and GP-Si-Luc targeted bacteria through ABC sugar transporter, we constructed two types of bacterial mutants, i.e., a deletion mutant for delta-lamB (ΔlamB) and a deletion mutant for delta-malE (ΔmalE), followed by incubation with GP-Si-BPs or GP-Si-Luc. The Sanger sequencing data (Supplementary Notes) together with PCR data (Supplementary Fig. 14) verified the successful construction of ΔlamB and ΔmalE. As expected, we did not observe any fluorescent signals in ΔlamB or ΔmalE treated with GP-Si-BPs (Fig. 4d) or GP-Si-Luc (Fig. 4e). The results of flow cytometry were in agreement with CLSM images (Supplementary Fig. 15). Moreover, we further investigated whether different sugars (e.g., GP, glucose and maltotriose) would affect the uptake of GP-Si-BPs by ABC sugar transporters. As revealed in Supplementary Fig. 16, the uptake of GP-Si-BPs was inhibited by an excess of GP or maltotriose rather than glucose. As previously reported, the ABC sugar transporters allowed the transportation of not only maltose but also linearly (1-4)-glucosidically

linked glucose polymers, such as maltodextrins, amylose and starch, as well as (1-4)-glucosidically linked cyclodextrins[57,58]. These results together proved that the uptake mechanism of Trojan BLI probes into bacteria was indeed through ABC sugar transporter pathway.

Finally, we demonstrated the good specificity of GP-Si-BPs and GP-Si-Luc for bacteria over mammalian cells. Typically, human blood samples spiked with MDR $E. coli$ or MRSA were incubated with GP-Si-BPs (0.06 mM), GP-Si-Luc (0.06 mM) or GP-Si-BPs + GP-Si-Luc (0.06 mM) for 2.5 h and then washed with PBS buffer. As indicated in Supplementary Fig. 17, green and red fluorescence signals were only observed in bacterial cells rather than in human blood cells. These results demonstrated that GP-Si-BPs or GP-Si-Luc were scarcely internalized by mammalian cells due to the absence of ABC sugar transporters in mammalian cells.

To determine the detection limit, we imaged a concentration series of MRSA and MDR $E. coli$ in PBS buffer incubated with Trojan BLI probes (Fig. 4f). Typically, bacteria at concentrations as low as ~10$^6$ CFU produced a detectable signal. On the basis of this, we further demonstrated the presented Trojan horse BLI strategy enabled ex vivo bioluminescence imaging of ten kinds of pathogens in the vitreous collected from patients with bacterial endophthalmitis, and the imaged pathogens were $Enterococcus faecalis$, $Klebsiella pneumoniae$, $Streptococcus pneumoniae$, $Neisseria meningitidis$, $Haemophilus influenzae$, $Streptococcus pyogenes$, Vancomycin (Van)-resistant $Enterococcus$, multidrug-resistant $Staphylococcus aureus$ (MRSA), $Moraxella catarrhalis$, and $Salmonella para-typhi A$, as revealed in Fig. 4g. These pathogens were confirmed by bacterial culturing, also provided by the Shanghai Eye, Ear, Nose and Throat Hospital, Fudan University. These results implied the potential of Trojan horse BLI strategy in the clinical bacterial diagnostics.

### Bioluminescence imaging of bacterial nephritis in mice by using Trojan horse strategy

Next, we testified the feasibility of the proposed strategy on bioluminescent imaging of natural bacteria in deep tissues. To this end, we first constructed a proof-of-concept model of $S. aureus$-induced nephritis in mice (Supplementary Fig. 18a). The actual $S. aureus$ concentration at the infection site during imaging was ~1.0 × 10$^8$ CFU. We found that the strongest bioluminescence signals appeared in the infected sites only in GP-Si-BPs + GP-Si-Luc groups (Fig. 5a). Quantitatively, the signal intensity in GP-Si-BPs + GP-Si-Luc groups was ~3-fold stronger than that in GP-Si-BPs + Si-Luc groups. We could also observe the similar results in the ex vivo imaging of the harvested kidney tissues (Supplementary Fig. 19). In addition, as low as ~1.0 × 10$^6$ CFU $S. aureus$ in kidney could be discriminated by the developed Trojan strategy (Fig. 5b), and such sensitivity should be sufficient for many in vivo scenarios. To demonstrate the selective imaging of the developed Trojan BLI strategy against bacterial nephritis over other nephritis, we next constructed glycerin-caused colitis in mice (Supplementary Fig. 18b). As expected, we found that the bioluminescent signals in $S. aureus$ nephritis-bearing mice treated with GP-Si-BPs + GP-Si-Luc were significantly brighter than the control and glycerin nephritis-bearing mice treated with GP-Si-BPs + GP-Si-Luc ($p < 0.001$) (Fig. 5c). For further comparison, we also utilized another self-luminescence reagent of luminol to image $S. aureus$ nephritis-bearing mice (Supplementary Fig. 18c) or glycerin nephritis-bearing mice (Supplementary Fig. 18d). As previously reported, luminol could be catalyzed by myeloperoxidase (MPO) to produce luminescence[59,60]. Distinguished from the developed Trojan BLI strategy in the selective image of $S. aureus$ nephritis, luminol could produce strong luminescent signals in both $S. aureus$ nephritis-bearing mice and glycerin nephritis-bearing mice (Fig. 5d). These results together demonstrated that the developed Trojan BLI strategy could discriminate between bacterial nephritis and other nephritis. The high selectivity as well as the adjustable sensitivity of the developed strategy was ascribed to the selective internalization

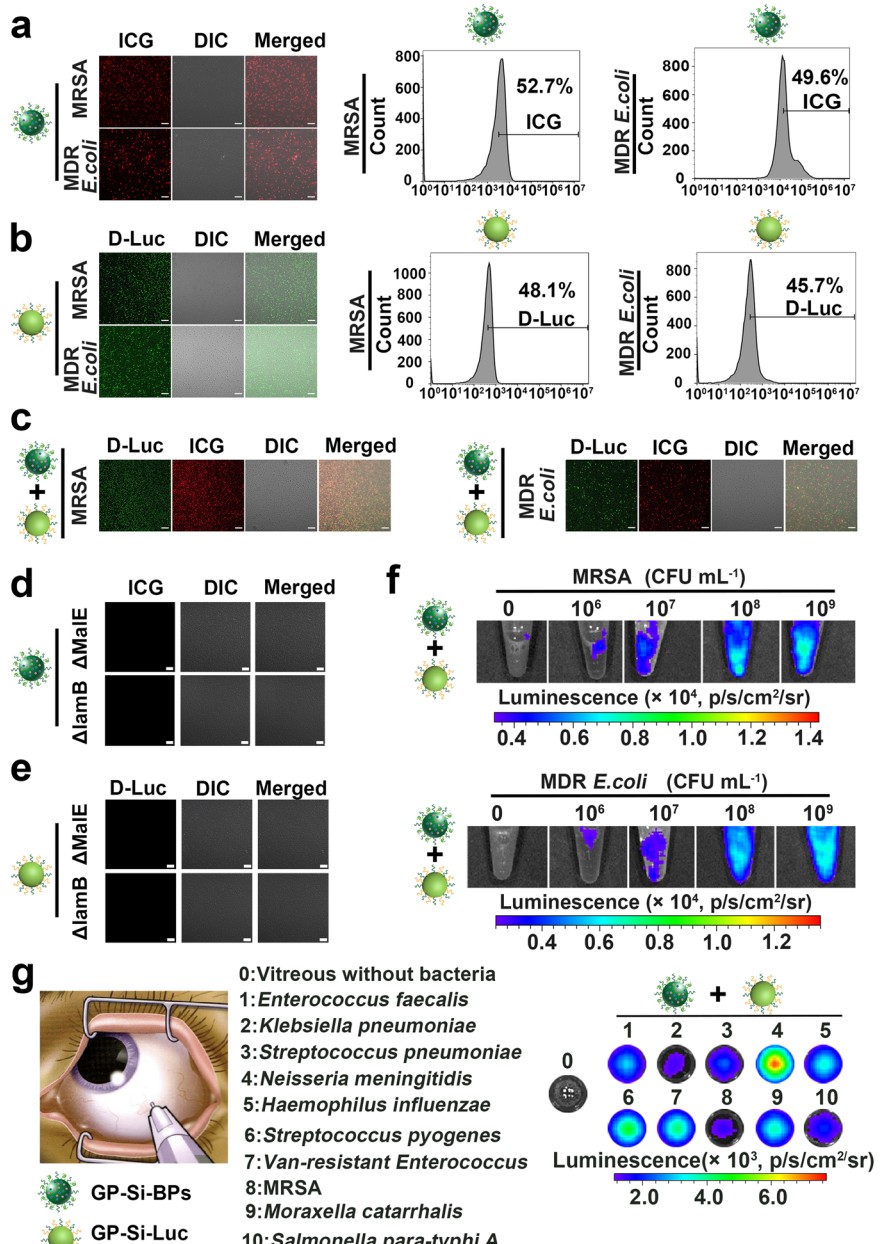

**Fig. 4 | Trojan BLI probes targeting diverse bacteria. a** Confocal fluorescence images of MRSA or MDR *E. coli* after incubation with 0.06 mM GP-Si-BPs and corresponding flow cytometry analysis of uptake rates. **b** Confocal fluorescence images of MRSA or MDR *E. coli* after incubation with 0.06 mM GP-Si-Luc and corresponding flow cytometry analysis of uptake rates. **c** Confocal fluorescence images of MRSA or MDR *E. coli* after incubation with 0.06 mM GP-Si-Luc + GP-Si-BPs. **d, e** Confocal fluorescence images of bacteria mutants of ΔlamB and ΔmalE after incubation with 0.06 mM GP-Si-BPs (**d**) or GP-Si-Luc (**e**). The bacterial cell concentration is ~10⁷ CFU. Scale bar, 25 μm. **f** Bioluminescence images of PBS buffer containing MRSA or MDR *E. coli* with different concentrations after incubation with 0.06 mM GP-Si-Luc + GP-Si-BPs. **g** Ex vivo bioluminescence imaging of human vitreous containing *Enterococcus faecalis* (1), *Klebsiella pneumoniae* (2), *Streptococcus pneumoniae* (3), *Neisseria meningitidis* (4), *Haemophilus influenzae* (5), *Streptococcus pyogenes* (6), Vancomycin (Van)-resistant *Enterococcus* (7), multidrug-resistant *Staphylococcus aureus* (MRSA) (8), *Moraxella catarrhalis* (9), or *Salmonella para-typhi A* (10) collected from patients with bacterial endophthalmitis. The vitreous without bacteria (0) from a healthy volunteer was set as the control. All imaging experiments were repeated three times with similar results.

of Trojan BLI probes into the bacterial cells. In addition, we systematically compared the previously reported GP and Ce6 modified SiNPs (GP-Ce6-SiNPs) system (simply excitation of Ce6)[39], Trojan BLI system, and ICG-based NIR emission imaging in the imaging of *S. aureus* nephritis (Supplementary Fig. 20). Although GP-Ce6-SiNPs system and ICG-based NIR emission imaging allowed the detection of bacterial uptake of nanoagents in the kidney owing to the specific bacterial homing capability of GP ligands, Trojan BLI strategy featured 3.75 times higher signal-to-noise ratios than that obtained by GP-Ce6-SiNPs system and 3.46 times higher signal-to-noise ratios than that obtained

by ICG-based NIR emission imaging. Taken together, we used Trojan BLI rather than Ce6 or ICG-based emission imaging in the detection of *S. aureus* nephritis owing to the high-contrast imaging of the Trojan BLI system.

**Bioluminescence imaging of bacterial colitis in mice by using Trojan horse strategy**

To confirm the generality of the developed strategy in the imaging of natural bacteria in deep tissues, we constructed another proof-of-concept model of STm-induced colitis in mice (Supplementary

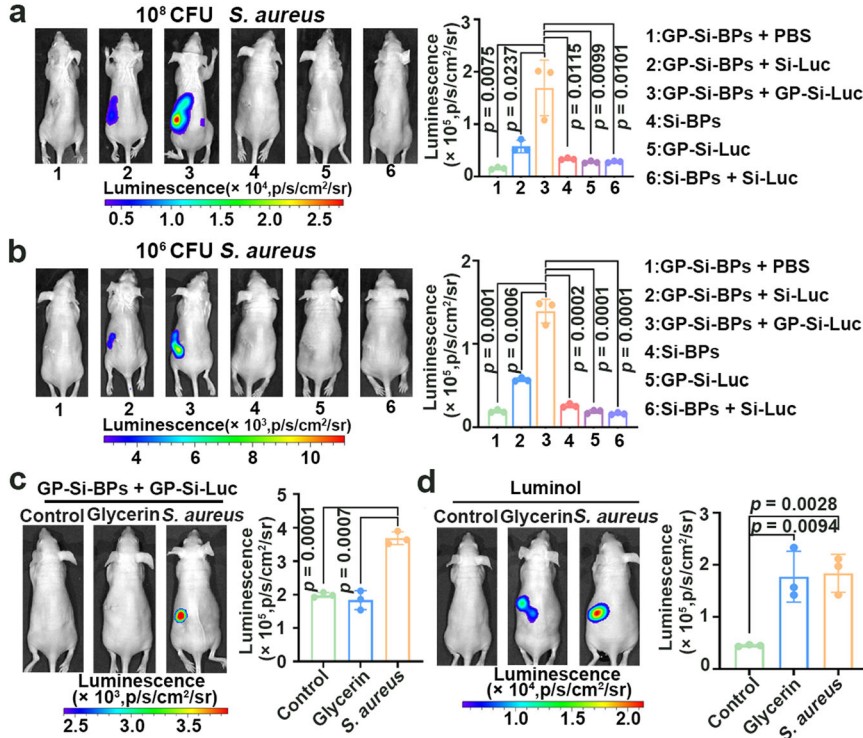

**Fig. 5 | In vivo imaging in mouse models of nephritis based on the proposed Trojan horse strategy. a** Bioluminescence imaging of *S. aureus* (~$1.0 \times 10^8$ CFU)-induced nephritis in mice with different treatments as indicated. The infected mice were injected with GP-Si-BPs + PBS, GP-Si-BPs + Si-Luc, GP-Si-BPs + GP-Si-Luc, Si-BPs, GP-Si-Luc and Si-BPs + Si-Luc at the same dose respectively (mean ± SD, $n = 3$). **b** Bioluminescence imaging of *S. aureus* (~$1.0 \times 10^6$ CFU)-induced nephritis in mice with different treatments as indicated. The infected mice were injected with GP-Si-BPs + PBS, GP-Si-BPs + Si-Luc, GP-Si-BPs + GP-Si-BPs, Si-BPs, GP-Si-Luc and Si-BPs + Si-Luc at the same dose, respectively (mean ± SD, $n = 3$). **c** Bioluminescence imaging

of health mice (control), glycerin nephritis-bearing mice and *S. aureus* nephritis-bearing mice by using the Trojan BLI probes (mean ± SD, $n = 3$). **d** Luminescence imaging of health mice (control), glycerin nephritis-bearing mice and *S. aureus* nephritis-bearing mice by intraperitoneal administration of luminol (282 mM, 200 μL) (mean ± SD, $n = 3$). All imaging experiments were repeated three times with similar results. Statistical analysis was performed using a one-way ANOVA analysis. Error bars represent the standard deviation obtained from three independent measurements. Source data are provided as a Source data file.

Fig. 21a). The actual amount of STm at the infection site during imaging was determined as ~$1.0 \times 10^9$ CFU. As revealed in Fig. 6a, we observed much stronger bioluminescent signals in GP-Si-BPs + GP-Si-Luc groups than in other groups ($p < 0.0001$). Consistent results were observed in the ex vivo imaging of the isolated intestinal tissues (Fig. 6b). To testify the selectivity of the developed Trojan BLI strategy against bacterial colitis over other colitis, we constructed dextran sulfate sodium (DSS)-induced colitis in mice (Female, 6–8 weeks old, $n = 3$) (Supplementary Fig. 21b). The DSS colitis-bearing mice were treated with GP-Si-BPs + GP-Si-Luc under the identical conditions. Indeed, the bioluminescent signals in STm groups treated with GP-Si-BPs + GP-Si-Luc were significantly stronger than the control and DSS groups treated with GP-Si-BPs + GP-Si-Luc ($p < 0.001$) (Fig. 6c). We also employed luminol to image STm colitis-bearing mice (Supplementary Fig. 21c) or DSS colitis-bearing mice (Supplementary Fig. 21d) for further comparison. As expected, compared with ignorable signals in control groups, we observed strong luminescent signals in both STm colitis-bearing mice treated with luminol ($p < 0.0001$) and DSS colitis-bearing mice treated with luminol ($p < 0.01$) (Fig. 6d). These results together demonstrated that the developed Trojan BLI strategy could discriminate between bacterial colitis and other colitis. We also demonstrated that the developed Trojan BLI strategy enabled long-term and real-time image of STm colitis in mice (Fig. 6e). Typically, we could still observe distinct signals at 1 h after administration of GP-Si-Luc. This observed persistent luminescence together with the good specificity was attributed to the selective accumulation of Trojan BLI probes into the bacterial cells in the infected intestinal tissues. Also, we systematically compared the GP-Ce6-SiNPs system (simply excitation of Ce6)[42], Trojan BLI strategy,

and ICG-based NIR emission imaging in the imaging of STm colitis (Supplementary Fig. 22). As expected, Trojan BLI strategy featured 4.05 times higher signal-to-noise ratios than that obtained by GP-Ce6-SiNPs system and 3.45 times higher signal-to-noise ratios than that obtained by ICG-based NIR emission imaging. Thereby, we used Trojan BLI instead of Ce6 or ICG-based emission imaging in the detection of STm colitis owing to the high-contrast imaging of the Trojan BLI system.

## Antibacterial activity of the developed strategy

Besides for the bioluminescence imaging of bacteria, the constructed GP-Si-BPs also possessed the potential to inactive bacteria due to the photothermal effects originated from ICG in GP-Si-BPs. As revealed in SEM images of bacteria (e.g., *E. coli*, *S. aureus*) (Supplementary Fig. 23a), we observed wrinkled or lysed bacterial cells only in the groups in which the bacteria were incubated with GP-Si-BPs (0.06 mM) for 2.5 h and then irradiated by an 808-nm laser (1.0 W cm$^{-2}$) for 5 min. We have shown the solid evidence highlighting the superiority of GP-Si-BPs in direct comparison to the clinically used antibiotics (e.g., vancomycin (Van), ampicillin (Ampi)) (Supplementary Fig. 23b). Overall, the presented strategy has two distinct advantages over antibiotics in killing bacteria: (1) GP-Si-BPs can kill Gram-positive bacteria as well as Gram-negative bacteria, while vancomycin is only workable for Gram-positive bacteria (e.g., *S. aureus*, *M. luteus* and MRSA); (2) the developed strategy shows dominant antibacterial rates (e.g., >90%) during a short-time treatment (e.g., 2 h and 30 min), while vancomycin or ampicillin even at 15 μg mL$^{-1}$ displays inferior antibacterial rates even the treating time is up to 7 h (e.g., <65%) (Supplementary Figs. 23c &

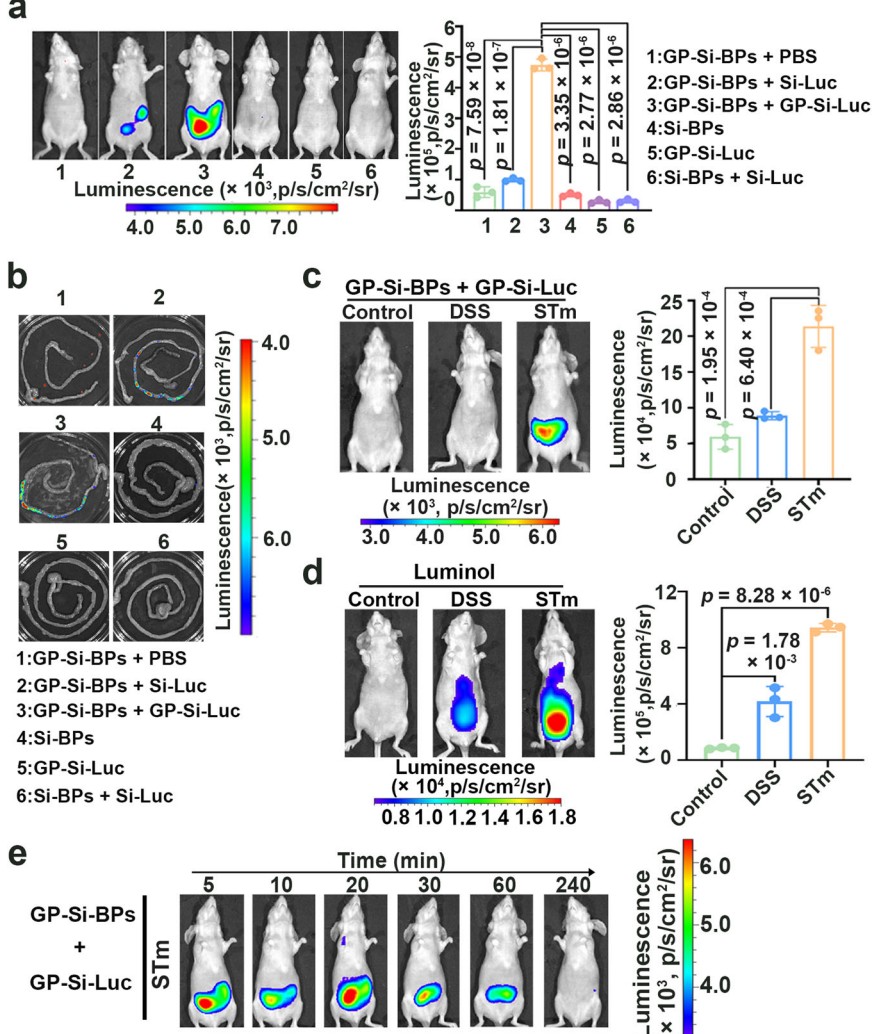

**Fig. 6 | In vivo imaging in mouse models of colitis based on the proposed Trojan horse strategy. a** Bioluminescence imaging of STm (~1.0 × 10⁹ CFU)-induced colitis in mice with different treatments as indicated. The infected mice were injected with GP-Si-BPs + PBS, GP-Si-BPs + Si-Luc, GP-Si-BPs + GP-Si-Luc, Si-BPs, GP-Si-Luc and Si-BPs + Si-Luc at the same dose respectively (mean ± SD, *n* = 3). **b** Corresponding ex vivo luminescent images of intestinal tissues isolated from the STm colitis-bearing mice with different treatments as indicated. **c** Bioluminescence imaging of health mice (control), DSS colitis-bearing mice and STm colitis-bearing mice based on the Trojan BLI probes (mean ± SD, *n* = 3). **d** Luminescence imaging of health mice (control), DSS colitis-bearing mice and STm colitis-bearing mice by intra-peritoneal administration of luminol (282 mM, 200 μL) (mean ± SD, *n* = 3). **e** Time-lapse bioluminescent imaging of STm (~1.0 × 10⁹ CFU)-induced colitis in mice based on the Trojan BLI probes. All imaging experiments were repeated three times with similar results. Statistical analysis was performed using a one-way ANOVA analysis. Error bars represent the standard deviation obtained from three independent measurements. Source data are provided as a Source data file.

23d). These results convincingly demonstrate the utility of such treatments to kill bacteria over the antibiotics.

To evaluate the antibacterial ability of the developed strategy in vivo, we examined its photothermal efficacy in the proof-of-concept model of bacterial nephritis in mice (Supplementary Fig. 24). We also excised the infected kidney tissues after the treatment, followed by homogenization, and culturing the homogenates on plates. As revealed in Fig. 7a, b, we found the sporadic colonies existed in GP-Si-BPs + 808-nm laser group after 7 days of treatment. Quantitatively, the in vivo antibacterial rate of GP-Si-BPs under laser irradiation against *S. aureus* was calculated to be 96.0%. In line with the agar plate experiments, hematoxylin-eosin (H&E) staining data (Fig. 7c) showed that the clear tissue texture and no cell necrosis was found only in GP-Si-BPs + 808-nm laser group. The PPT effects contributed to the high antibacterial rates. We used an IR thermal imaging camera to record the in vivo photothermal images. As displayed in Fig. 7d, the rapid temperature rising only occurred in GP-Si-BPs + 808-nm laser group. Particularly, the kidney temperature could enhance to 52 °C after 5 min irradiation in this group. Meanwhile, the developed Trojan BLI strategy

also allowed the dynamic visualization of anti-bacterial effect in mice. Experimentally, mice were intraperitoneally injected with the same dose of GP-Si-Luc after each irradiation, followed by bioluminescence imaging to evaluate the curative effect. As displayed in Fig. 7e, the bioluminescence signals gradually decreased as the treatment progressed. As expected, the change trend in bioluminescence signals was consistent with the change trend in bacterial count after treatment. Through the quantitative analysis, we found that bacterial concentration-dependent BLI signals were an important indicator for detection and treatment. These therapeutic data together proved that the adaptable antibacterial ability of the developed strategy in vivo.

## Toxicity assessment

Ultimately, we systematically investigated the toxicity of GP-Si-BPs and GP-Si-Luc. As revealed in Supplementary Fig. 25, the morphologies of HEK-293T, mREC, HeLa or MCF-7 cells hardly changed when they were incubated with GP-Si-BPs + GP-Si-Luc (0.06 mM). Also, the cell viability of these cells remained above 80% by using the methyl thiazolyl tetrazolium (MTT) assays. These results suggest the feeble cytotoxicity of

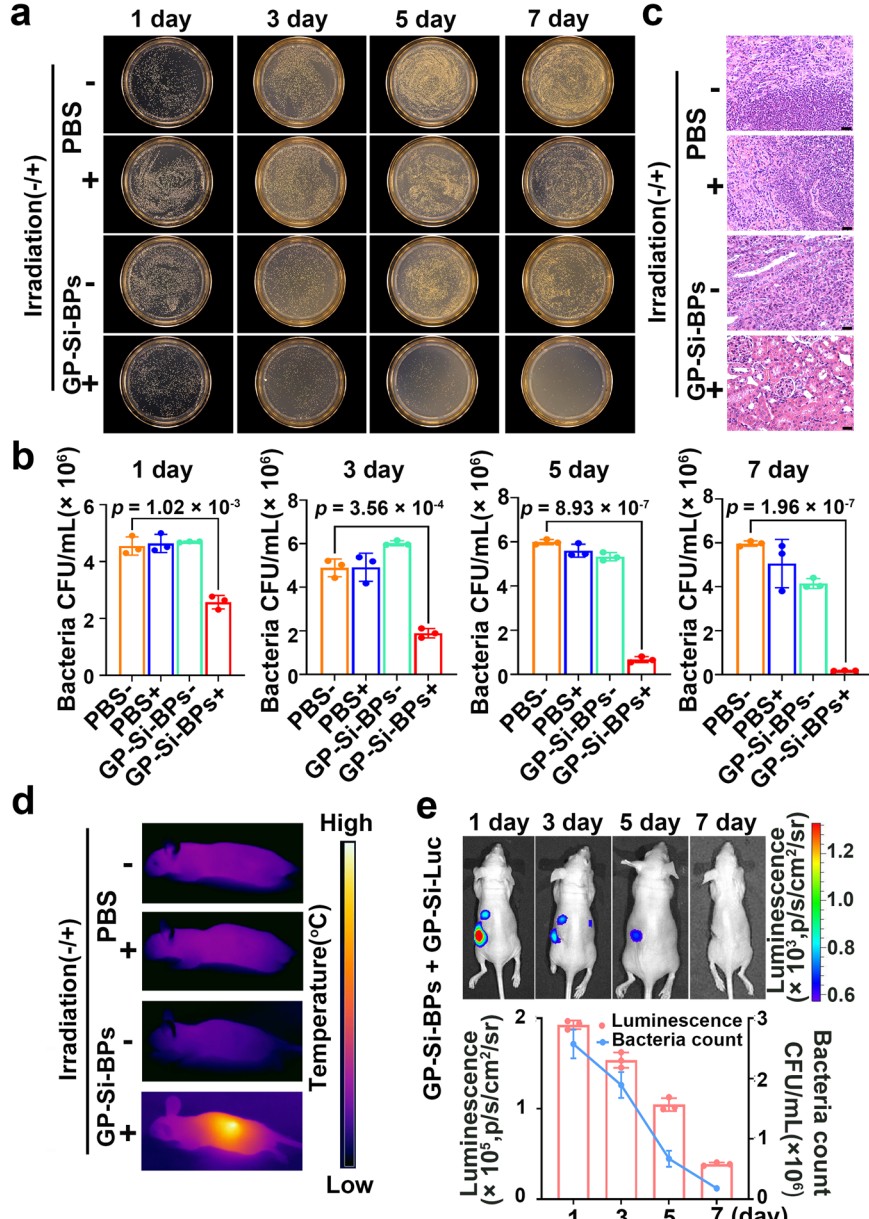

**Fig. 7 | In vivo antibacterial activity in bacterial nephritis-bearing mice based on Trojan horse strategy. a, b** Homogenates of infected kidney in nephritis-bearing mice treated by PBS or GP-Si-BPs with or without irradiation (with irradiation of 808-nm laser, 1.0 W cm⁻², 5 min) at 7-day post-injection cultured on the solid LB agar (**a**) and corresponding quantification of bacterial colonization (mean ± SD, $n = 3$ biologically independent samples) (**b**). **c** Corresponding histological images of *S. aureus*-infected kidney tissues of mice with different treatments as indicated. Scale bar, 25 μm. **d**, Representative photothermal images of *S. aureus* nephritis-bearing mice treated with PBS or GP-Si-BPs with or without irradiation (with irradiation of 808-nm laser, 1.0 W cm⁻², 5 min) at 7-day post-injection.

**e** Bioluminescence imaging of *S. aureus* nephritis-bearing mice at different treating time points using Trojan BLI probes and the corresponding relationship between the change in bioluminescence intensity over time and the change in bacterial count over time by quantitative comparison. After each irradiation, mice were intraperitoneally injected with the same dose of GP-Si-Luc for imaging to evaluate the curative effect (mean ± SD). All imaging experiments were repeated three times with similar results. Statistical analysis was performed using a one-way ANOVA analysis. Error bars represent the standard deviation obtained from three independent measurements. Source data are provided as a Source data file.

GP-Si-BPs and GP-Si-Luc. We used blood biochemistry and routine blood analysis to test the systematic toxicity of GP-Si-BPs and GP-Si-Luc at the tested dose. As shown in Supplementary Tables 2 & 3, compared with the PBS group, all biochemical indicators in GP-Si-BPs group or GP-Si-Luc group were within the normal range. Moreover, we also performed H&E (Supplementary Fig. 26a) and Masson's trichrome staining (Supplementary Fig. 26b) of major organs harvested from the mice after 1 day of intravenous injection of GP-Si-BPs (0.06 mM) or GP-Si-Luc (0.06 mM). Together, we found no obvious histopathological abnormalities in the sections of the resected organs, indicating the

neglectable in vivo toxicity of GP-Si-BPs or GP-Si-Luc. As further revealed in ex vivo fluorescence imaging of major organs resected from healthy mice after intravenous injection of GP-Si-BPs (Supplementary Fig. 27a), bright fluorescence signals were mainly observed in liver and kidney rather than other organs at 2 h post-injection, and would be totally disappeared at 48 h post-injection. Furthermore, strong fluorescence signals were observed in the urine samples collected from the healthy mice after 24 h post-injection of GP-Si-BPs (Supplementary Fig. 27b). These results suggested that the constructed probes could be eliminated from the body.

## Discussion

Herein, we developed a Trojan horse strategy to selectively and robustly deliver luciferase and luciferin into various natural bacteria, visualizing the bacterial distribution in vivo with bioluminescence. Bioluminescence, as a promising non-invasive self-luminescence imaging modality, refers to the photon emission of the exothermic oxidation of the corresponding luciferin catalyzed by luciferase[18–25]. Despite the great prospects of bioluminescence, it has hardly allowed the imaging of commensal and pathogenic bacteria due to the following limitations: (1) the endogenous bioluminescence was only produced by some specific engineered bacteria, and was not applicable to most natural bacteria; (2) the production of exogenous bioluminescence required the participation of bacterial ATP, which was obtained by bacterial lysis[29–34]. An easier way to visualize various natural bacteria in vivo with bioluminescence may be to selectively delivery bioluminescent reporter into bacterial cells, directly consuming the ATP inside the bacteria, yet present technical challenges due to the electrostatic charge or size barriers imposed by the unique bacterial plasma membrane and cell wall. Intriguingly, Trojan horse antibiotic strategies developed in 1980s could deliver siderophores-conjugated antibiotics into bacterial cytoplasm through the bacteria iron importers[35–41]. Inspired by this, we proposed Trojan horse BLI strategy.

Distinguished from the Trojan horse antibiotic strategies, in our design, the bioluminescent indicators were loaded onto GP-modified SiNPs, being transported into bacteria through ABC sugar transporters. Of note, SiNPs, as the vectors, have been extensively employed in various bioapplications due to their low/no toxicity[58–64]. As previously reported, SiNPs could be biodegraded and be renally cleared without distinct toxicity in vivo[65,66]. For instance, toxicity assessments in *Caenorhabditis elegans*, silkworm and even in non-human primate-animal models (e.g., cynomolgus macaques) have demonstrated that the internalized SiNPs would not affect the morphology, physiology or innate immune functions of animals, suggesting benign biocompatibility of SiNPs in live organisms[67–69]. On the basis of this, small-size SiNPs have received the Food and Drug Administration (FDA)-approved investigational new drug approval for the first-in-human clinical trial in January 2011 (i.e., NCT01266096, NCT02106598)[70,71]. As such, we selected the SiNPs as the vehicles to deliver bioluminescent indicators. The bioluminescent indicators included luciferase, and two organic dyes, Cy5 and ICG. As a result, NIR emission can be achieved using a dual resonance energy transfer relay process that combines BRET between luciferase and Cy5, and FRET between Cy5 and ICG. Thereby, the developed strategy has the potential to image bacteria in deep tissues in specific clinical settings. Notwithstanding, the combination of luciferase, Cy5, and ICG requires relatively complicated works for sample preparation and quantitative analysis, and this significantly lowers the impact of this work. In the following study, we would test simpler system.

On the other aspect, pioneering research over the past three decades has elucidated the mechanisms, by which the unique carbon source of bacterial cells such as maltodextrins, amylose and starch, as well as (1-4)-glucosidically linked cyclodextrins could be selectively internalized into bacterial cells through the bacteria-specific ABC sugar transporter[46–53]. However, these insights have not yielded ABC transporter-based strategies successful in delivering bioluminescent cargo into bacterial cells. To fill this gap, we leveraged bacteria-specific ABC sugar transporters to selectively internalize luciferase and luciferin by hitchhiking them on GP molecules with dextrose equivalent of 4.0-7.0. The synthesized Trojan BLI probes were selectively internalized into both Gram-positive bacteria (e.g., *S. aureus*, clinically-derived MRSA) and Gram-negative bacteria (e.g., STm, clinically-derived MDR *E. coli*) with a high uptake rate up to 50%. We showed its ability in ex vivo imaging of bacterial endophthalmitis in human. We showed its selective image against bacterial nephritis and bacterial colitis in mice over other types of nephritis and colitis in mice. Besides, we demonstrated the developed strategy allowed photothermal killing of nearly 95% of the antibiotic-resistant bacteria in vitro and in vivo. We anticipate the proposed Trojan BLI strategy could catalyze the delivery of various substances into bacterial cells, allowing the development of various imaging or therapeutic approaches, with bioluminescence imaging of bacteria reported in this work serving as an initial proof of principle.

## Methods

### Ethical statement

The research protocol using human blood samples was approved by the Ethics Committee of Soochow University. Samples were provided following written informed consent. The authors declare that all human blood experiments were conducted in strict accordance with relevant laws and institutional guidelines. Clinical samples were provided by the Eye Bank of the Eye, Ear, Nose and Throat Hospital, Fudan University with the approval of the hospital ethics committee (EEN-TIRB-2017-06-07-01). All experiments were conducted in accordance with the Declaration of Helsinki and in compliance with Chinese law.

### Fabrication of Trojan BLI probes

Upon 7-day and 365-nm irradiation at room temperature on the mixture of $C_6H_{17}NO_3Si$ and 1,8-naphthalimide, the small-size fluorescent SiNPs were prepared. Next, the resultant solution was purified by centrifugation at 3381 x g for 15 min and dialysis (MWCO, 1000, Spectra/Pro) to remove unreacted reagents. The final products of SiNPs were collected through evaporation and stored at 4 °C for the following experiments. To obtain GP-modified SiNPs (GP-SiNPs), the mixture of SiNPs solution (200 μL, 0.08 mM) and GP solution (100 μL, 0.56 mM) was continuously stirred at 70 °C for 6 h, followed by the addition of 0.01 mg of NaBH₄ for reacting for another 12 h at room temperature. The resultant solution was purified by Nanosep centrifugal devices (MW cutoff, 3 kDa; Millipore) through centrifugation at 5283 x g for 15 min to remove the unreacted GP molecules. To further load Cy5 and ICG molecules, Cy5 solution (2 μL, 0.56 mM) and ICG solution (12 μL, 0.56 mM) were added into the above prepared GP-SiNPs solution and stirred at room temperature overnight. Similarly, the resultant solution was purified by Nanosep centrifugal devices (MW cutoff, 3 kDa; Millipore) through centrifugation at 6010 x g for 10 min to remove the unloaded Cy5 and ICG molecules. Afterwards, amine-containing luciferase solution (0.1 mL, 0.06 mM), EDC solution (2 μL, 10 mM) and NHS solution (10 μL, 17 mM) were added into the above-prepared solution for reaction over night at room temperature to obtain GP-Si-BPs. Also, the resultant solution was purified by Nanosep centrifugal devices (MW cutoff, 100 kDa; Millipore) through centrifugation at 5283 x g for 15 min. Analogously, GP-Si-Luc was also synthesized by the addition of D-luciferin (0.1 mL, 0.30 mM), EDC solution (2 μL, 10 mM) and NHS solution (10 μL, 17 mM) to GP-SiNPs solution over night at room temperature followed by the identical purified protocol.

### Characterizations of Trojan BLI probes

The morphology and size of Trojan BLI probes were characterized by using a transmission electronic microscopy (TEM, Philips CM 200) with 200 kV. The UV-vis absorption spectra of Trojan BLI probes were recorded by using a 750 UV-vis near-infrared spectrophotometer (Perkin-Elmer lambda). The photoluminescence spectra of Trojan BLI probes were recorded by using a spectro-fluorimeter (HORIBA JOBIN YVON FLUORMAX-4). The bioluminescence spectra of Trojan BLI probes were recorded by using a HITACHI Fluorescence Spectrometer F-4700 instrument. Dynamic light scattering (DLS) and Zeta potentials of Trojan BLI probes were measured by using a Delsa™ nano submicron particle size and Zeta potential particle analyzer (Beckman Coulter, Inc). Fluorescence imaging of bacteria in vitro were performed by using a confocal laser scanning microscope (CLSM, Leica, TCS-SP5 II). Ex vivo

and in vivo fluorescence and bioluminescence images were obtained by an in vivo optical imaging system (IVIS Lumina III). Luminescent signals of GP-Si-BPs in different solutions were examined using an IVIS Lumina III imaging system. To this end, GP-Si-BPs at 0.06 mM was mixed with ATP at different concentrations in a black 96-well plate. After thorough mixing, the plate was immediately put into the imaging system to acquire luminescent signals. Similarly, the attenuated luminescence of GP-Si-BPs was quantified after incubation with an ATP inhibitor, DCC, in the presence of GP-Si-Luc.

## Bacterial culture

Clinically-derived multidrug-resistant *Staphylococcus aureus* (MRSA) and multidrug-resistant *Escherichia coli* (MDR *E. coli*) were isolated from patients with keratitis and were supplied by the Eye Bank of the Eye, Ear, Nose and Throat Hospital, Fudan University, under the approval of the hospital ethics committee (EENTIRB-2017-06-07-01). *Escherichia coli* (ATCC 11303) and *Staphylococcus aureus* were bought from American type culture collection (ATCC). *Micrococcus luteus* (BNCC 102589), *Pseudomonas aeruginosa* (BNCC 125486) and *Salmonella typhimurium* (BNCC 108207) were purchased from BeNa Culture Collection (BNCC, Shanghai, China). The LB medium was purchased from Sangon Biotech (Shanghai) Co., Ltd. To culture the bacteria, we first dissolved the lyophilized powder of strains in the LB liquid medium, then coated the bacteria liquid onto the LB plate medium and cultured them in the incubator at 37 °C for 12 h. After that, we picked out a single colony from the plate and cultured it in the LB liquid medium in the incubator at 250 rpm and 37 °C. We collected the bacterial cells at the exponential growth phase. The concentration of bacteria was detected by measuring the optical density (OD) at 600 nm. The number of bacterial colonies was counted by a colony counting instrument (Czone 8). Experiments were conducted according to the Declaration of Helsinki and in compliance with Chinese law.

## In vitro imaging of bacteria

The 20 μL of purified and re-suspended bacterial suspension ($1.0 \times 10^7$ CFU) was incubated with GP-Si-BPs (0.06 mM, 200 μL), GP-Si-Luc (0.06 mM, 200 μL), GP-Si-BPs (0.06 mM, 200 μL) + GP-Si-Luc (0.06 mM, 200 μL), Si-BPs (0.06 mM, 200 μL), Si-Luc (0.06 mM, 200 μL) and Si-BPs (0.06 mM, 200 μL) + Si-Luc (0.06 mM, 200 μL) for 2.5 h in a shaking incubator (250 rpm) at 37 °C. The bacteria were harvested by centrifuging the mixture at 3381 x g for 5 min in Eppendorf (EP) tubes. The resulting bacteria were re-suspended and washed with PBS for three times. Then 10 μL of the washed bacteria solution was transferred onto a microscope slide covered by a coverslip, and then imaged by a confocal laser scanning microscope (CLSM, Leica, TCSSP5 II) with 30% power of diode laser. All fluorescence images were captured by CLSM with a × 64 oil-immersion objective and taken under the same optical conditions, and the same brightness and contrast was applied to the images by the microscope automatically. The processing and analysis of ROI was performed by the commercial image analysis software (Leica Application Suite Advanced Fluorescence Lite). Moreover, the appearance of GP-Si-BPs in the bacterial cells were confirmed by TEM (Philips CM 200).

## In vivo imaging of bacteria

All animal experimental procedures were performed according to the protocol approved by the animal care committee of Soochow University. The housing conditions for the mice were 25 °C and 65% humidity adjusted by the ventilation equipment and air filtration system. To construct STm-induced colitis in mice, female nude mice (SPF grade, 6–8 weeks old) were treated with 100 μL of streptomycin solution (200 mg mL⁻¹) prior to the orally administration of STm. At the second day after infection, the mice were intravenously injected with GP-Si-BPs or Si-BPs (0.06 mM, 200 μL). At 6 h post-injection of probes, the treated mice were intraperitoneally injected with GP-Si-Luc

or Si-Luc (0.06 mM, 200 μL), followed by BLI by using an in vivo optical imaging system (IVIS Lumina III, exposure time = 5 min, f/stop = 1, binning = 8, FOV = 21.8 cm) at 15 min post-injection of substrates. For a comparison, the infected mice were intraperitoneally injected with luminol (282 mM, 200 μL), followed by BLI by using an in vivo optical imaging system (IVIS Lumina III, exposure time = 5 min, f/stop = 1, binning = 8, FOV = 21.8 cm) at 10 min post-injection. We used the intestinal tissue harvesting, homogenization, and culturing with CFU count to determine the actual number of bacteria at the infection sites during imaging. The actual concentration of STm at the infection site during imaging was ~$1.0 \times 10^9$ CFU. To test the selectivity of the proposed strategy, we constructed DSS-induced colitis in mice (Female, 6–8 weeks old, $n = 3$). The DSS colitis-bearing mice were treated with GP-Si-BPs + GP-Si-Luc (0.06 mM, 200 μL) or luminol (282 mM, 200 μL) for luminescence imaging (IVIS Lumina III, exposure time = 5 min, f/stop = 1, binning = 8, FOV = 21.8 cm) in the same manner as STm colitis-bearing mice mentioned above. To construct *S. aureus*-induced nephritis in mice, 25 μL of *S. aureus* were in situ injected into the kidney of the nude mice (Female, 6–8 weeks old, $n = 3$). At 12 h post-injection, the infected mice were intravenously injected with 200 μL of 0.06 mM GP-Si-BPs or Si-BPs. Six hours later, these mice were intraperitoneally injected with GP-Si-Luc or Si-Luc (0.06 mM, 200 μL), followed by in vivo imaging by using an optical imaging system (IVIS Lumina III, exposure time = 5 min, f/stop = 1, binning = 8, FOV = 21.8 cm) at 15 min post-injection. The actual *S. aureus* concentration at the infection site during imaging was ~$1.0 \times 10^8$ CFU, which was determined through kidney tissue harvesting, homogenization, and culturing with CFU count as mentioned above. Also, to test the selectivity of the proposed strategy against bacterial nephritis over other nephritis, we constructed glycerin-caused nephritis in mice (Female, 6–8 weeks old, $n = 3$). The glycerin nephritis-bearing mice were treated with GP-Si-BPs + GP-Si-Luc (0.06 mM, 200 μL) or luminol (282 mM, 200 μL) for luminescence imaging (IVIS Lumina III, exposure time = 5 min, f/stop = 1, binning = 8, FOV = 21.8 cm) in the same manner as DSS nephritis -bearing mice mentioned above.

## In vitro antibacterial assays

*S. aureus* or *E. coli* were respectively incubated with PBS or GP-Si-BPs, followed by the irradiation of 808-nm laser (808-nm laser:1.0 W cm⁻², 5 min). The morphology of bacteria after treatment was characterized by using SEM (FEI Quanta 200 F). *S. aureus*, *E. coli*, *M. luteus*, *P. aeruginosa*, MRSA and MDR *E. coli* were respectively treated with PBS, SiNPs, Si-BPs and GP-Si-BPs followed by the irradiation of 808-nm laser (808-nm laser, 1.0 W cm⁻², 5 min). The antibacterial rate in vitro was obtained based on the bacteria colonies on the agar plates. The antibacterial rate was calculated according to Eq. (1):

$$\text{Antibacterial rate}\,(\%) = (N_{control} - N_{experiment})/N_{control} \times 100\%, \quad (1)$$

where "$N_{control}$" and "$N_{experiment}$" represent bacterial counts (CFU mL⁻¹) in the control groups of "PBS" and other experimental groups (experiment), respectively.

**In vivo antibacterial assays.** We used the *S. aureus*-induced nephritis in mice to evaluate the antibacterial ability of the developed strategy in vivo. To construct the model, *S. aureus* (25 μL, ~$1.1 \times 10^6$ CFU) were in situ injected into the left kidney of the mice (Female, 6–8 weeks old, $n = 3$). At 6 h post-injection, these mice were intravenously injected with 200 μL of 0.06 mM GP-Si-BPs or PBS buffer on day 1, day 3, day 5 and day 7 respectively. Six hours after each drug injection, the infected sites were irradiated with or without 808-nm laser (1.0 W cm⁻², 5 min). After photothermal therapy, the infected kidney issues were extracted, followed by homogenization, and culturing the homogenates on plates. The corresponding antibacterial rate was calculated based on

Eq. (1). Meanwhile, the infected kidney tissues after treatments were fixed in the 4% PFA solution for the following H&E staining.

**Cellular experiments in vitro.** Human embryonic kidney 293 T cells (HEK-293T cells), human cervical cells (HeLa cells), human breast cancer cells (MCF-7 cells) and mouse retinal endothelial cells (mREC cells), cultured in the Dulbecco's modified Eagle's medium with high glucose (H-DMEM), were purchased from Shanghai Zhong Qiao Xin Zhou Biotechnology Co., Ltd (China). All above-mentioned media were supplemented with 10% heat-inactivated fetal bovine serum (FBS) and 1% relevant antibiotics (100 µg mL$^{-1}$ streptomycin and 100 U mL$^{-1}$ penicillin). All cell lines were cultured at 37 °C in a 5% $CO_2$ incubator with the humidified atmosphere.

### Clinical studies
Human blood samples were provided by a healthy volunteer following written informed consent. The study protocols using human blood samples were approved by the ethics committee of Soochow University. The authors state that all human blood experiments were performed in strict accordance with the relevant laws and institutional guidelines. Ten patients with bacterial endophthalmitis who were undergoing vitreous surgery, uncontaminated non-diluted vitreous fluid samples (0.1 ml) were collected in a syringe with a 30 G needle during diagnostic pars plana vitrectomy (PPV). Immediately after collection, the sample was transferred into a pre-sterilized microfuge tube and used for imaging. These clinical samples also were supplied by the Eye Bank of the Eye, Ear, Nose and Throat Hospital, Fudan University, under the approval of the hospital ethics committee (EENTIRB-2017-06-07-01).

### Statistics & reproducibility
For statistical significance testing, we used a one-way ANOVA analysis or the paired two-tailed $t$-test. The statistical analysis was performed by using the software of Origin or GraphPad Prism. Error bars represent the standard deviation obtained from three independent measurements. Region of interest (ROI) was employed for quantitative assessments of fluorescence intensity, which was calculated by the commercial image analysis software (Leica Application Suite Advanced Fluorescence Lite 2. 6. 0, LAS AF Lite 2. 6. 0).

Group sizes for experiments were chosen on the basis of prior experience and literature precedence, so that sufficient numbers were used to ensure reproducibility and determine standard deviations. The number of animals was at least 3. For each sample, two technical replicates were carried out. If there was 20% or greater variation between technical replicates, an additional two technical replicates were carried out. All experiments were carried out with at least 3 replicate samples for each experimental group, and we confirmed all attempts at replication were successful. No data were excluded from the analyses. Samples were allocated into experimental groups at random. All the data collection and analysis were from blinded with randomized samples.

### Reporting summary
Further information on research design is available in the Nature Portfolio Reporting Summary linked to this article.

## Data availability
The data that support the findings of this study are available within the paper and its supplementary information. Source data are provided with this paper.

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

## Acknowledgements

We thank Prof. Shuit-Tong Lee (Soochow University, China), Prof. Chunhai Fan (Shanghai Jiao Tong University, China) and Dr. Fei Peng (Harvard University, USA) for their general help and valuable suggestions. Y. H. discloses support for the research described in this study from National Natural Science Foundation of China [grant number 21825402], Program for Jiangsu Specially Appointed Professors to Professor Yao He, a project funded by the Priority Academic Program Development of Jiangsu Higher Education Institutions (PAPD), 111 Project and Collaborative Innovation Center of Suzhou Nano Science and Technology (NANO-CIC). H. Y. W. discloses support for the research described in this study from National Natural Science Foundation of China [grant number 22074101] and Natural Science Foundation of Jiangsu Province of China [grant number BK20191417]. J. X. H. discloses support for the research described in this study from National Natural Science Foundation of China [grant number 81970766, 82171102], the Shanghai Innovation Development Program [grant number 2020-RGZN-02033] and the Shanghai Key Clinical Research Program [grant number SHDC2020CR3052B]. B. S. discloses support for the research described in this study from National Natural Science Foundation of China [grant number 22204116] and Natural Science Foundation of Jiangsu Province of China [grant number BK20200851]. B. B. C. discloses support for the research described in this study from National Natural Science Foundation of China [grant number 22204117] and China Postdoctoral Science Foundation [grant number 2021M692347].

## Author contributions

Q.Z., B.S., J.X.H., Y.N.X., Y.M.Y., H.Y.W., and Y.H. conceived and designed the research. Q.Z. and B.S. carried out most of experiments and analyzed the data. Y.M.Y., J.J., W.J.C., J.P.L., J.L.D., H.T.C., and B.B.C performed additional experiments and characterizations. Q.Z., Y.N.X., J.X.H., H.Y.W and Y.H wrote the manuscript.

## Competing interests

The authors declare no competing interests.
