## [Peer Review File · Nature Communications]

Reviewers' comments:

Reviewer #1 (Remarks to the Author):

In this study, authors report a bioluminescence imaging (BLI) of bacteria as a means of pathogen detection. Denoted as a Trojan horse strategy to deliver luciferase and luciferin into various bacteria for in vivo visualization without any genetic modification of the natural bacteria, this manuscript suggests a promising non-invasive imaging method. In their previous work, silicon nanoparticles modified with glucose polymer (GP) can be readily internalized into various bacteria through the ABC transporter pathway, which is a key concept for bacterial detection in this study as well. Although authors discriminate the use of BRET and FRET for effective imaging over tissue autofluorescence to detect in vivo NIR signal compared to their previous study using Ce6, the GP-Si-BPs and GP-Si-Luc dual nanoparticles system does not sufficiently show technical advancement and strategic novelty to be fitted for high standard profile in Nature Communications. Particularly, it is still quite doubtful the reason why authors use BLI system rather than simply excite ICG for in vivo imaging as it is widely used for non-destructive in vivo imaging. Technically, using the combination of luciferase, Cy5, and ICG for bacterial visualization is one of the good side works for verifying another potential imaging methods. However, its use in the bacterial infection seems not to be a good target, as it doesn't have specific bacterial homing capability and it requires relatively complicated works for sample preparation and quantitative analysis. Overall, authors report extensive work on in vivo bioluminescence imaging method as a Trojan horse strategy for bacterial visualization and detection with systematic experiments that support main idea and following studies. However, scientific novelty and advancement based their findings are not significantly sufficient to be published in Nature Communications. Therefore, the reviewer recommends this manuscript against the publication in this journal. However, this study still contains lots of interesting findings, particularly as a branch study of ABC sugar transporter-based bacterial detection, thus is considered to be transferred to more specific journals such as Communications Materials or Microsystems & Nanoengineering after revision of following comments:

Major points:

1. Authors claim to use BLI rather than ICG-based NIR emission imaging. However, it is hardly understood in the present manuscript. This should be more clearly described as bacterial uptake of GP-Si-BPs is also clearly detectable under NIR irradiation.
2. It is unclear how much amount of dye molecules (Cy5 and ICG) are adsorbed onto the GP-SiNPs. Authors described electrostatic adsorption of the dye molecules at known quantity. Is it assumed that all the dye molecules were completely adsorbed onto the nanoparticles? Otherwise, authors should quantitatively declare the dye contents loaded in the nanoparticles, as it is very important factor in following studies of BLI and photothermal therapy.
3. In Figure 2e, the bioluminescence signal slightly decreases at longer period of time. However, it is not clearly explained why relatively stable luminescence signal is detected in bioluminescence. Since the

emission mechanism of the luciferase-based bioluminescence is different with that of fluorescence, the stability cannot be compared in this way. Continuous 808 nm irradiation to ICG is absolutely responsible for potential bleaching and quenching effect for reduced emission, while enzymatic reaction between luciferin and luciferase might be stably maintained as it isn't affected by external energy (e.g., light irradiation). Although the emission intensity of the FL goes down under longer excitation, minimal emission can be overwhelming than bioluminescence. In this regard, the emission intensity scales on Figure 2e should be correct in BLI and FL for quantitative comparison.

4. Authors showed the BLI is still significantly higher than the FL, which also has higher tissue penetration (Figure 2f). However, the initial emission (depth = 0 mm) of BLI and FL should be also added, and the tissue thickness-dependent intensity decrease should be normalized to the intensity at depth = 0 mm accordingly in Figure 2f.

5. There are many microorganisms in the body, including pathogenic and non-pathogenic bacteria. Is there any strategic methods to distinguish those bacteria for selective antibacterial treatment?

6. Authors claimed the superior antibacterial efficacy in vitro (Figure 6). However, the reason why the GP-Si-BPs have great inactivation of bacteria has not been described. Also, bacterial concentration-dependent BLI signals are an important indicator for detection and treatment. It would be great if authors add more quantitative analysis and detection strategy on bacterial infection based on the BLI of GP-Si-BPs.

Minor correction needed:

1. In Figure 1a, schematic representation of each component such as bacteria in the colon domain is not clearly visible. It would be better to visualize clearer in the schematics.

2. TEM images of both GP-Si-BPs and GP-Si-Lic are not clear in Figure 1d and e. It would be great to add high-resolution TEM images of both nanoparticles in the supplementary information, as the size, shape, morphology of nanoparticles are important parameter to be considered in bacterial consumption. Also, it would be nice to add bare silicon nanoparticles prior to chemical conjugation of GP, Cy5, ICG, and either luciferase or luciferin.

3. In Figure 1f, schematic labels are not clearly visible in the present form. It would be great to revise with letter labels rather than schematics.

4. Stability of luciferase and luciferin after Si NP conjugation?

Reviewer #2 (Remarks to the Author):

The concept that is described in the manuscript is interesting, novel, creative and promising. To my best knowledge, this concept in bacterial imaging has never been published. The authors describe a thorough testing of this method in vitro and in vivo, which also has a large potential value.

However, in its current state the manuscript lacks several basic elements needed for transparent research, that allows for a qualitative review process. I will address these basic elements below. Subsequently, a more detail review of the first 8 pages is given. After addressing the mentioned basic elements by the authors, a further detailed review of the rest of the manuscript would be necessary.

- The result section, including the figures, encompasses 20 pages. For the majority it does not only describe actual results of the current research, but also introduction, methods and hypothesis. Moreover, is not clearly marked where theoretical hypothesis transfers to actual results. This should be substantially restructured and shortened.
- Of specific results and figures, it is unclear how these were generated. A couple of examples:
 - o Lines 133-134: it is unclear how these exact numbers were derived. Was DLS used for this? In this case, suppl fig 1 should show how these numbers were generated. Moreover, at the small TEM images of fig 1d-e, using the scale bar, the particles seem smaller than 5.6 nm, so how this related than to the supposed DLS data is not completely clear to me and should be explained. Moreover, in Fig 1 d and e, it is not clear to me how based on this image it is concluded that the particles are indeed spherical.
 - o Figure 1f shows one of the key elements of the article. However, it is not described how this figure should be interpreted.
 - o Suppl fig 3a: a linear correlation cannot be claimed by the number and spacing of the measured points.
 - o Suppl fig 3b: it is not clear how the data for this figure was generated. Methods are lacking.
- Specific methods, such as ATP assay and DCC, are not included in the methods. Please include all methods.
- The figures are cluttered and not well structured. It is difficult to get the intended message from the figures. In footnotes of figures, it should be explained what can be seen in the figure. For example, in fig 1f and suppl fig 2, this is not clear. On the other side, repetition of the methods is not specifically necessary in footnotes.
- The research is of additive value if the researcher intends a use of the probe in specific clinical settings in vivo. In the results and discussion this is suggested, e.g., by testing a probe on human clinical samples (in this case vitreous), and when referring to FDA approval, respectively. This should be more thoroughly addressed and discussed, including limitations.

- In extension of a supposed clinical use of probe, stability testing is lacking.
- The introduction and discussion should include a thorough description and consideration of the BRET/FRET concept.
- Check English grammar.
- Whether the research described in this manuscript is novel enough compared to earlier published work by the same authors in Nat com, is questionable. However, the described concept in itself, is indeed very novel.

Detailed review of the first 8 pages:

- The title claims “Deep organ” imaging. This is not described in the manuscript and could be considered misleading.
- The advantage of BLI imaging without genetical engineering and BLI imaging without a need of damaging the bacterial cells, by using a Trojan horse technique could be clearer addressed in the last sentence of the abstract.
- The probe is tested in clinical materials. The term “human testing” is misleading.
- Chemical yields are lacking in synthesis.
- Line 29: add “in vitro”.
- Line 46-48: “A key ... host organism” the meaning of this sentence was not clear for me.
- Line 49 and 60: please avoid the “and so forth”.
- Line 50-52: infection is not an inflammation symptom, which is implied by the formulation of this sentence.

- Line 54-57: near infrared imaging has not such a large autofluorescence problem as described in this sentence. The certainty of the sentence should be tuned down/nuanced.
- Line 68-70: first, the...pathogenic bacteria” This is a non-sensical sentence. It was already stated that non-genetically engineered bacteria lack BLI. Therefore, it is logical that BLI is naturally not present in commensal and pathogenic bacteria. Furthermore, if BLI is not present, it is not “difficult to visualize endogenous BLI systems” when not present, but that is intrinsically impossible.
- Line 70-72: This is a non-sensical sentence. Please rephrase where it is more clearly addressed that the availability of extracellular ATP is a major limitation for imaging live bacteria.
- Line 82-83: please explain and/or reference why α (1-4)-glucosidically linked glucose polymers (dextrose equivalent of 4.0~7.0)-linked nanoparticles are used.
- Line 84: please replace “kinds” by “species” or “strains”.
- Line 138: “human-derived” could be rephrased, e.g., as clinically-derived.
- Line 138: Abbreviations, such as MDR and MRSA, should be written out in full.
- Line 145: surface and morphology are not “identical”, but could be described as comparable.
- Line 146: “election” should be “electron”?
- Figure 1a: this illustrates the hypothesis, but is not a result. It should be uncoupled from the results section.
- Figure 1b: the added value of this figure is unclear.
- Figure 1f shows one of the key elements of the article. However, it is not described how this figure should be interpreted.

- Figure 1g: the control including GP-Si-Luc with Si-BP is lacking. Please add this. Moreover, the lysed positive control with GP-Si-BPs and GP-Si-Luc might be of additive value and should be added.

苏州大学

复旦大学附属眼耳鼻喉科医院

EYE & ENT HOSPITAL OF FUDAN UNIVERSITY

上海市五官科医院

Address: 83 Road Fenyang, Xuhui District, Shanghai, 200031, China

Telephone: +86-021-64377134

Fax: +86-021-64377151

Website: www.fdeent.org

Point-by-Point Response to Reviewer Comments

Reviewer #1 (Remarks to the Author):

In this study, authors report a bioluminescence imaging (BLI) of bacteria as a means of pathogen detection. Denoted as a Trojan horse strategy to deliver luciferase and luciferin into various bacteria for *in vivo* visualization without any genetic modification of the natural bacteria, this manuscript suggests a promising non-invasive imaging method. In their previous work, silicon nanoparticles modified with glucose polymer (GP) can be readily internalized into various bacteria through the ABC transporter pathway, which is a key concept for bacterial detection in this study as well. Although authors discriminate the use of BRET and FRET for effective imaging over tissue autofluorescence to detect *in vivo* NIR signal compared to their previous study using Ce6, the GP-Si-BPs and GP-Si-Luc dual nanoparticles system does not sufficiently show technical advancement and strategic novelty to be fitted for high standard profile in Nature Communications. Particularly, it is still quite doubtful the reason why authors use BLI system rather than simply excite ICG for *in vivo* imaging as it is widely used for non-destructive *in vivo* imaging. Technically, using the combination of luciferase, Cy5, and ICG for bacterial visualization is one of the good side works for verifying another potential imaging methods. However, its use in the bacterial infection seems not to be a good target, as it doesn't have specific bacterial homing capability and it requires relatively complicated works for sample preparation and quantitative analysis. Overall, authors report extensive work on *in vivo* bioluminescence imaging method as a Trojan horse strategy for bacterial visualization and detection with systematic experiments that support main idea and following studies. However, scientific novelty and advancement based their findings are not significantly sufficient to be published in Nature Communications. Therefore, the reviewer recommends this manuscript against the publication in this journal.

Summary of response: We thank the reviewer for his/her thoughtful comments. Accordingly, the point-by-point responses to the comments made by Reviewer #1 are given below.

- First, the advance and novelty of the paper has been further investigated based on a deeper dive into the literature. To simplify the discussion, please see below a table summarizing the features of our previous works based on glucose polymer (GP)-modified nanoagents. We agree with Reviewer #1's point that one key concept in these works is that GP-modified nanoagents can be readily internalized into various bacteria through the ABC transporter pathway, as summarized in **Supplementary Table 1**. Typically, in our first paper (*Nat. Commun.* **10**, 4057 (2019)), we for the first time used the nanoagents made of GP and Ce6 modified SiNPs for fluorescence imaging and photodynamic killing of various bacteria in superficial skin tissues. On the basis of this work, we developed the first example of photoacoustic imaging and photodynamic and photothermal killing of various bacteria in gut and tumour tissues by using GP and Ce6 modified gold nanoparticles (AuNPs) (e.g., *Nat. Commun.* **13**, 1255 (2022)). In addition to bacterial detection and inactivation, we used GP-modified nanoagents to construct the first demonstration of a bacteria-based drug delivery system for cancer treatment (e.g., *Nat. Commun.* **13**, 5127 (2022)). Apparently, these works published in Nature Communications have moved forwards step by step, sufficiently showing technical advancement and strategic novelty. Likewise, this submitted work is a big step forwards from the previous works. We present **the first demonstration of bioluminescence imaging and photothermal killing of various bacteria in deep**

tissues by using GP-modified nanoprobe. Specifically, the developed nanoprobe is totally different from what we have reported before. Different from other BLI methods for bacterial detection (e.g., *Sci. Adv.* **7**, eaaz9857 (2021); *Nat. Protoc.* **3**, 629–636 (2008); *Nat. Biotechnol.* **22**, 313–320 (2004); etc.), **this is the first paper to use BLI to visualize various natural bacteria *in vivo*, based on the delivery of bioluminescence reports to bacterial cells that directly consume ATP inside the bacteria.** Taken together, we believe the presented work fully shows technical advancement and strategic novelty to be fitted for a high standard profile in Nature Communications.

Supplementary Table 1. Comparison of the features of glucose polymer (GP)-modified agents.

Refs.	Features		
	Agents composition	Mechanisms of entry into bacteria	Functions
Nat. Commun. 10 , 4057 (2019)	GP and chlorin e6 (Ce6) modified silicon nanoparticles (SiNPs)	ABC transporter pathway	Fluorescence imaging and photodynamic killing of gram-negative and gram-positive bacteria
Nat. Commun. 13 , 1255 (2022)	GP and Ce6 modified gold nanoparticles (AuNPs)	ABC transporter pathway	Photoacoustic imaging and photodynamic and photothermal killing of gram-negative and gram-positive bacteria
Nat. Commun. 13 , 5127 (2022)	GP and indocyanine green (ICG) modified SiNPs	ABC transporter pathway	A bacteria-based drug delivery system for glioblastoma photothermal immunotherapy
Angew. Chem. Int. Ed. 61 , e202208422 (2022)	GP and ICG modified SiNPs, AuNPs or carbon dots (CDs)	ABC transporter pathway	A bacteria-based drug delivery system for photothermal programmable destruction of deep tumor tissues
This work	GP, luciferase, Cy5 and ICG-modified SiNPs, GP and D-luciferin-modified SiNPs	ABC transporter pathway	Bioluminescence imaging and photothermal killing of gram-negative and gram-positive bacteria

• Second, we have provided more details and experimental data to explain why we use the BLI system rather than simply excitation of ICG for *in vivo* imaging. **Please see the details in the Response to**

your Comment 1. We agree with Reviewer #1's point that ICG is widely used for nondestructive *in vivo* imaging. Indeed, the BLI system can directly achieve NIR emission from ICG under excitation for *in vivo* imaging. However, fluorescence imaging might cause relatively strong background autofluorescence from biological tissues, resulting in a relatively poor signal-to-background ratio (e.g., *Nat. Commun.* **10**, 1058 (2019); *Nat. Commun.* **3**, 1193 (2012); *etc.*). Through systematically comparative experiments (**Supplementary Fig. 20 & Supplementary Fig. 22**), we clearly demonstrate that the current BLI system features higher signal-to-noise ratios (e.g., ~3.46 times in the detection of bacterial nephritis and ~3.45 times in the detection of bacterial colitis) than ICG-based NIR emission imaging. Therefore, we used the BLI system rather than simply exciting ICG for *in vivo* imaging.

- Third, in the BLI system, the purpose of the combination of luciferase, Cy5, and ICG is to leverage BRET and FRET, enabling NIR bacterial visualization, similar to other reports (e.g., *Nat. Commun.* **11**, 4192 (2020); *Nat. Commun.* **3**, 1193 (2012); *etc.*). Technically, GPs in the BLI system have specific bacterial homing capability, as supported by other reports (e.g., *Nat. Commun.* **10**, 4057 (2019); *Nat. Commun.* **13**, 1255 (2022); *Nat. Mater.* **10**, 602–607 (2011); *Nat. Commun.* **11**, 1250 (2020); *etc.*). We agree with Reviewer #1's point that the combination of luciferase, Cy5, and ICG requires relatively complicated work for sample preparation and quantitative analysis. In the following study, we will test a simpler system. We have discussed this limitation in the revised manuscript. Notwithstanding, the related quantitative details have been provided in the manuscript. **Please see the details in the Response to your Comment 2.** In brief, the amounts of luciferase, Cy5, and ICG loaded onto SiNPs can be quantified based on the corresponding calibration absorption curves (**Supplementary Fig. 2**). To obtain the equivalent dose, the detected absorbance of luciferase, Cy5, and ICG should be kept the same among groups. Similar sample preparation and quantitative analysis methods have also been employed in other reports (e.g., *Nat. Commun.* **10**, 4057 (2019); *Nat. Commun.* **13**, 1255 (2022); *Nat. Commun.* **13**, 5127 (2022); *Angew. Chem. Int. Ed.* **61**, e202208422 (2022)).

Overall, as Reviewer #1 pointed out, we “**report extensive work on *in vivo* bioluminescence imaging method as a Trojan horse strategy for bacterial visualization and detection with systematic experiments that support main idea and following studies**”. We believe the scientific novelty and advancement based our findings are significantly sufficient to be published in Nature Communications, which is also supported by Reviewer #2's comments: “**The concept that is described in the manuscript is interesting, novel, creative and promising. To my best knowledge, this concept in bacterial imaging has never been published. The authors describe a thorough testing of this method *in vitro* and *in vivo*, which also has a large potential value.**”.

Location of changes: Supplementary Table 1, Supplementary Fig. 20, Supplementary Fig. 22, Paragraph 2 on Page 4, Paragraph 1 on Page 17 and Paragraph 1 on Page 20.

(1) Authors claim to use BLI rather than ICG-based NIR emission imaging. However, it is hardly understood in the present manuscript. This should be more clearly described as bacterial uptake of GP-Si-BPs is also clearly detectable under NIR irradiation.

Response: Thank you very much for Reviewer #1's suggestion. Accordingly, we have added the

苏州大学

复旦大学附属眼耳鼻喉科医院

EYE & ENT HOSPITAL OF FUDAN UNIVERSITY

上海市五官科医院

Address: 83 Road Fenyang, Xuhui District, Shanghai, 200031, China

Telephone: +86-021-64377134

Fax: +86-021-64377151

Website: www.fdeent.org

related description that the bacterial uptake of GP-Si-BPs could be detectable under NIR irradiation. We designed a series of experiments and added related discussions to explain why we used the BLI system rather than ICG-based NIR emission imaging. Several papers have used the NIR bioluminescence generated from BRET-FRET effects instead of simply excitation of NIR fluorescent dye for *in vivo* imaging applications such as lymph-node mapping and cancer imaging ascribed to the high signal-to-noise ratio and the high spatial resolution of NIR bioluminescence imaging (e.g., *Nat. Commun.* **11**, 4192 (2020); *Nat. Commun.* **3**, 1193 (2012); *etc.*).

To further help Reviewer #1 understand why we used the BLI system rather than Ce6 or ICG-based NIR emission imaging, we designed and performed a series of experiments to compare the GP-Ce6-SiNPs system (simplified excitation of Ce6), Trojan BLI system (GP-Si-BPs + GP-Si-Luc), and ICG-based NIR emission imaging (simplified excitation of ICG in GP-Si-BPs) (**Supplementary Fig. 20** and **Supplementary Fig. 22**). As revealed in **Supplementary Fig. 20**, the GP-Ce6-SiNPs system, ICG-based NIR emission imaging and Trojan BLI system all allowed the detection of bacterial uptake of nanoagents in the kidney owing to the specific bacterial homing capability of GP ligands, as pointed out by Reviewer #1. Comparatively, the Trojan BLI strategy featured 3.75 times higher signal-to-noise ratios than those obtained by the GP-Ce6-SiNPs system and 3.46 times higher signal-to-noise ratios than those obtained by ICG-based NIR emission imaging in the detection of bacterial nephritis. Similar results were also observed in the imaging of bacterial colitis in **Supplementary Fig. 22**, i.e., the Trojan BLI strategy featured 4.05 times higher signal-to-noise ratios than those obtained by the GP-Ce6-SiNPs system and 3.45 times higher signal-to-noise ratios than those obtained by ICG-based NIR emission imaging. Taken together, we used Trojan BLI rather than Ce6 or ICG-based emission imaging in the detection of various bacteria within deep tissues owing to the high-contrast imaging of the developed Trojan BLI system.

Supplementary Fig. 20. Comparison of GP-Ce6-SiNPs system (simply excitation of Ce6), Trojan BLI system (GP-Si-BPs + GP-Si-Luc), and ICG-based NIR emission imaging (simply excitation of ICG in GP-Si-BPs) in the imaging of *S. aureus*-induced nephritis. a, A scheme illustrating the optical imaging of *S. aureus*-induced nephritis in mice by using the GP-Ce6-SiNPs system, Trojan BLI system or ICG-based NIR emission imaging. b, Fluorescence imaging of healthy mice and *S. aureus* nephritis-bearing mice treated with or without GP-Ce6-SiNPs (excitation: 460 nm, emission: 670 nm). c, Fluorescence imaging of healthy mice and *S. aureus* nephritis-bearing mice treated with or without GP-Si-BPs (excitation: 780 nm, emission: 845 nm). d, Bioluminescence imaging of healthy mice and *S. aureus* nephritis-bearing mice treated with GP-Si-BPs + GP-Si-Luc or not (emission: 845 nm). e, Corresponding signal-to-noise ratios obtained by the GP-Ce6-SiNPs system, ICG-based NIR emission imaging and Trojan BLI system in the imaging of *S. aureus*-induced nephritis in mice. Statistical analysis was performed using one-way ANOVA. Error bars represent the standard deviation obtained from three independent measurements. Source data are provided as a source data file.

Supplementary Fig. 22. Comparison of GP-Ce6-SiNPs system (simply excitation of Ce6), Trojan BLI system (GP-Si-BPs + GP-Si-Luc), and ICG-based NIR emission imaging (simply excitation of ICG in GP-Si-BPs) in the imaging of STm-induced colitis. a, A scheme illustrating the luminescent imaging of STm-induced colitis in mice by using the GP-Ce6-SiNPs system, Trojan BLI system or ICG-based NIR emission imaging. **b,** Fluorescence imaging of healthy mice and STm colitis-bearing mice treated with GP-Ce6-SiNPs or not (excitation: 460 nm, emission: 670 nm). **c,** Fluorescence imaging of healthy mice and STm colitis-bearing mice treated with GP-Si-BPs or not (excitation: 780 nm, emission: 845 nm). **d,** Bioluminescence imaging of healthy mice and STm colitis-bearing mice treated with GP-Si-BPs + GP-Si-Luc or not (emission: 845 nm). **e,** Corresponding signal-to-noise ratios obtained by the GP-Ce6-SiNPs system, ICG-based NIR emission imaging and Trojan BLI system in the imaging of STm-induced colitis in mice. Statistical analysis was performed using one-way ANOVA. Error bars represent the standard deviation obtained from three independent measurements. Source data are provided as a source data file.

Location of changes: Supplementary Fig. 20, Supplementary Fig. 22, Paragraph 1 on Page 17 and Paragraph 1 on Page 20.

(2) It is unclear how much amount of dye molecules (Cy5 and ICG) are adsorbed onto the GP-SiNPs.

苏州大学

复旦大学附属眼耳鼻喉科医院

EYE & ENT HOSPITAL OF FUDAN UNIVERSITY

上海市五官科医院

Address: 83 Road Fenyang, Xuhui District, Shanghai, 200031, China

Telephone: +86-021-64377134

Fax: +86-021-64377151

Website: www.fdeent.org

Authors described electrostatic adsorption of the dye molecules at known quantity. Is it assumed that all the dye molecules were completely adsorbed onto the nanoparticles? Otherwise, authors should quantitatively declare the dye contents loaded in the nanoparticles, as it is very important factor in following studies of BLI and photothermal therapy.

Response: Following Reviewer #1's significant suggestion, we strictly quantified the amount of dye molecules (Cy5 and ICG) adsorbed onto the GP-SiNPs by the corresponding calibration absorption curves (**Supplementary Fig. 2**). Accordingly, the actual amount of Cy5 adsorbed onto the GP-SiNPs was ~ 0.0044 mM, and the actual amount of ICG adsorbed onto the GP-SiNPs was ~ 0.026 mM. To obtain the equivalent dose, the detected absorbance of Cy5 and ICG should be kept the same among groups. Similar sample preparation and quantitative analysis methods have also been employed in other reports (e.g., *Nat. Commun.* **10**, 4057 (2019); *Nat. Commun.* **13**, 1255 (2022); *Nat. Commun.* **13**, 5127 (2022); *Angew. Chem. Int. Ed.* **61**, e202208422 (2022)).

Location of changes: Paragraph 1 on Page 6.

(3) In Figure 2e, the bioluminescence signal slightly decreases at longer period of time. However, it is not clearly explained why relatively stable luminescence signal is detected in bioluminescence. Since the emission mechanism of the luciferase-based bioluminescence is different with that of fluorescence, the stability cannot be compared in this way. Continuous 808 nm irradiation to ICG is absolutely responsible for potential bleaching and quenching effect for reduced emission, while enzymatic reaction between luciferin and luciferase might be stably maintained as it isn't affected by external energy (e.g., light irradiation). Although the emission intensity of the FL goes down under longer excitation, minimal emission can be overwhelming than bioluminescence. In this regard, the emission intensity scales on Figure 2e should be correct in BLI and FL for quantitative comparison.

Response: We agree with Reviewer #1's comment that the emission mechanism of luciferase-based bioluminescence is different from that of fluorescence, and although the emission intensity of FL decreases under longer excitation, minimal emission can be more overwhelming than bioluminescence. Following Reviewer #1's suggestion, the time-dependent intensity decrease has been normalized to the intensity at time = 5 min accordingly in **Figure 3e**. Other reports use similar data processing (e.g., *Nat. Commun.* **11**, 4192 (2020)). As revealed in **Figure 3e**, the BLI system exhibits relatively good bioluminescence stability, retaining 84% of the normalized signal after 120 min because the enzymatic reaction between luciferin and luciferase is stably maintained, as it is not affected by external energy (e.g., light irradiation). In contrast, the normalized signal of ICG in GP-Si-BPs sharply declined by 26% under continuous 808-nm irradiation for 120 min because continuous 808-nm irradiation of ICG is responsible for the potential bleaching and quenching effect for reduced emission. Accordingly, the related discussion to explain why a relatively stable luminescence signal is detected in bioluminescence has been added to the revised manuscript.

Fig. 3e. Time-resolved bioluminescence and fluorescence (excitation: 780 nm) images and corresponding normalized signal for quantitative comparison of bioluminescence and fluorescence signal change. The normalized signal is defined as the ratio of the emission intensity detected at any time to the emission intensity detected at 5 minutes. Bioluminescence is produced by a mixture of GP-Si-BPs (0.06 mM), D-luciferin (150 μ M) and ATP (10 μ M). All imaging experiments were repeated three times with similar results. Error bars represent the standard deviation obtained from three independent measurements.

Location of changes: Fig. 3e, Paragraph 2 on Page 9.

(4) Authors showed the BLI is still significantly higher than the FL, which also has higher tissue penetration (Figure 2f). However, the initial emission (depth = 0 mm) of BLI and FL should be also added, and the tissue thickness-dependent intensity decrease should be normalized to the intensity at depth = 0 mm accordingly in Figure 2f.

Response: Following Reviewer #1's helpful suggestion, the initial emission (depth = 0 mm) of BLI and FL has been added, and the tissue thickness-dependent intensity decrease has been normalized to the intensity at depth = 0 mm accordingly in **Figure 3f**. As revealed in **Fig. 3f**, the normalized bioluminescence signal declined with increasing chicken breast tissue thickness ranging from 0 to 35 mm. Typically, when the thickness was up to 30 mm, the normalized bioluminescent intensity of GP-Si-BPs was still significantly higher than that of ICG-based NIR emission imaging ($p < 0.0001$).

Fig. 3f. Chicken breast tissue thickness-dependent bioluminescence and fluorescence (excitation: 780 nm) images and corresponding normalized signal for quantitative comparison of bioluminescence and fluorescence signal change. The normalized signal is defined as the ratio of the emission intensity detected at any tissue thickness to the emission intensity detected at 0 mm. Bioluminescence is produced by a mixture of GP-Si-BPs (0.06 mM), D-Luciferin (150 μ M) and ATP (10 μ M). All imaging experiments were repeated three times with similar results. Error bars represent the standard deviation obtained from three independent measurements.

Location of changes: Fig. 3f, Paragraph 2 on Page 9.

(5) There are many microorganisms in the body, including pathogenic and non-pathogenic bacteria. Is there any strategic methods to distinguish those bacteria for selective antibacterial treatment?

Response: Thanks a lot for Reviewer #1's helpful comment. In principle, antisense oligonucleotides (ASOs) can limit or eradicate the expression of specific genes by sequence-directed targeting of messenger RNA, which has the potential to achieve selective antibacterial treatment (e.g., *Nat. Biotechnol.* **19**, 360-364 (2001); *Nat. Biotechnol.* **15**, 751-753 (1997); *Nat. Biotechnol.* **16**, 355-358 (1998); etc.). Of note, the prerequisite for the binding of ASOs to their complementary genes is their delivery into bacterial cells, but pristine ASOs are unable to enter bacteria freely, which requires the assistance of vectors. However, current vehicles for delivering ASOs cannot specifically distinguish between bacterial cells and mammalian cells, greatly hindering their preclinical or clinical treatment of bacterial infections, especially for antibiotic-resistant bacteria (e.g., *Nat. Biotechnol.* **38**, 845-855 (2020); *Nat. Biotechnol.* **35**, 230-237 (2017); *Nat. Biotechnol.* **19**, 360-364 (2001); etc.). To address this issue, we set out to leverage bacteria-specific ABC transporters to selectively internalize ASOs by hitchhiking them on GP-SiNPs. Typically, GP and antisense peptide nucleic acid (asPNA)-modified SiNPs (GP-SiNPs-asPNA) are internalized into bacterial cells. The asPNA consists of two consequent blocks: Ec108*acpP* and Sau101*fmhB*. After being internalized into the intracellular volume of multidrug-resistant (MDR) *E. coli*, the Ec108*acpP* block hybridized to its complementary mRNA target, the messenger RNA of Ec108*acpP* (*macpP*), inhibiting the synthesis of fatty acids. Analogously, when GP-SiNPs-asPNA enters *S. aureus* cells, the Sau101*fmhB* block combines with the messenger RNA of Sau101*fmhB* (*mfmhB*), preventing the synthesis of peptidoglycan. As such, this strategy enables selective antibacterial treatment. This work is now in preparation.

苏州大学

复旦大学附属 眼耳鼻喉科医院

EYE & ENT HOSPITAL OF FUDAN UNIVERSITY

上海市五官科医院

Address: 83 Road Fenyang, Xuhui District, Shanghai, 200031, China

Telephone: +86-021-64377134

Fax: +86-021-64377151

Website: www.fdeent.org

(6) Authors claimed the superior antibacterial efficacy *in vitro* (Figure 6). However, the reason why the GP-Si-BPs have great inactivation of bacteria has not been described. Also, bacterial concentration-dependent BLI signals are an important indicator for detection and treatment. It would be great if authors add more quantitative analysis and detection strategy on bacterial infection based on the BLI of GP-Si-BPs.

Response: Thanks a lot for Reviewer #1's important comment. Accordingly, the reason why the GP-Si-BPs have great inactivation of bacteria has been described in the revised manuscript. Typically, in **Supplementary Fig. 23**, we have shown solid evidence highlighting the superiority of GP-Si-BPs in direct comparison to clinically used antibiotics (e.g., vancomycin (Van) and ampicillin (Ampi)). Overall, the presented strategy has two distinct advantages over antibiotics in killing bacteria: (1) GP-Si-BPs can kill gram-positive bacteria (e.g., *S. aureus*, *M. luteus* and MRSA) as well as gram-negative bacteria (e.g., *E. coli*, *P. aeruginosa* and MDR *E. coli*) due to the broad-spectrum antibacterial properties of photothermal effects originated from ICG in GP-Si-BPs, while vancomycin is only workable for gram-positive bacteria (e.g., *S. aureus*, *M. luteus* and MRSA); (2) the developed strategy shows dominant antibacterial rates (e.g., ~95.9% to *S. aureus*, ~94.0% to *P. aeruginosa*, ~90.4% to *E. coli*, ~91.8% to *M. luteus*, ~96.0% to MDR *E. coli*, and ~95.8% to MRSA) during a short-time treatment (e.g., 2 hours and 30 mins) owing to the high antibacterial efficacy of photothermal effects originated from ICG in GP-Si-BPs, while vancomycin or ampicillin even at 15 $\mu\text{g}/\text{mL}$ displays inferior antibacterial rates even the treating time is up to 7 hours (e.g., vancomycin: ~44.3% to *S. aureus*, ~50.4% to *M. luteus* and ~45.5% to MRSA; ampicillin: ~52.7% to *S. aureus*, ~48.4% to *E. coli*, ~62.5% to *M. luteus*). These results convincingly demonstrate the utility of such treatments to kill bacteria over antibiotics. Similar results have also been observed in other reports (e.g., *Nat. Commun.* **13**, 1255 (2022)).

On the other hand, following Reviewer #1's helpful suggestion, we added more quantitative analysis and detection strategy on bacterial infection based on the BLI of GP-Si-BPs in **Figure 7e**. Experimentally, *S. aureus* nephritis-bearing mice (female, 6 - 8 weeks old, $n = 3$) were intravenously injected with 200 μL of 0.06 mM GP-Si-BPs on days 1, 3, 5 and 7, and photothermal treatment (PTT) was performed under 808-nm laser irradiation 6 hours after each drug injection. The photothermal treatment lasted for 5 minutes. After each irradiation, mice were intraperitoneally injected with GP-Si-Luc for imaging to assay the therapeutic effect *in vivo*. Meanwhile, we excised the infected kidney tissues after the treatment, followed by homogenization, and cultured the homogenates on plates. As displayed in the new **Figure 7e**, the bioluminescence signals gradually decreased as the treatment progressed. As expected, the change trend in bioluminescence signals was consistent with the change trend in bacterial count after treatment. Through quantitative analysis, we found that bacterial concentration-dependent BLI signals are an important indicator for detection and treatment.

Fig. 7e. Bioluminescence imaging of *S. aureus* nephritis-bearing mice at different treatment time points using Trojan BLI probes and the corresponding relationship between the change in bioluminescence intensity over time and the change in bacterial count over time by quantitative comparison. After each irradiation, mice were intraperitoneally injected with the same dose of GP-Si-Luc for imaging to evaluate the curative effect. All imaging experiments were repeated three times with similar results. Error bars represent the standard deviation obtained from three independent measurements.

Location of changes: Fig. 7e, Paragraph 2 on Page 21, Paragraph 1 on Page 22, Paragraph 2 on Page 22.

Minor correction needed:

1. In Figure 1a, schematic representation of each component such as bacteria in the colon domain is not clearly visible. It would be better to visualize clearer in the schematics.

Response: Following Reviewer #1's helpful suggestion, we have revised the schematic diagram in **Figure 1** to enlarge the details to ensure that the schematic representation of each component, such as bacteria in the colon domain, is clearly visible.

Fig. 1. Schematic design of ABC sugar transporter enabling selective delivery of bioluminescent nanoprobes into gram-positive bacteria and gram-negative bacteria to visualize various natural bacteria *in vivo* with bioluminescence by directly consuming the ATP inside the bacteria. The nanoprobes are made of GP, Cy5, ICG and luciferase-modified silicon nanoparticles (SiNPs) (GP-Si-BPs) and GP, D-luciferin-modified SiNPs (GP-Si-Luc).

Location of changes: Fig. 1.

...

2. TEM images of both GP-Si-BPs and GP-Si-Luc are not clear in Figure 1d and e. It would be great to add high-resolution TEM images of both nanoparticles in the supplementary information, as the size, shape, morphology of nanoparticles are important parameter to be considered in bacterial consumption. Also, it would be nice to add bare silicon nanoparticles prior to chemical conjugation of GP, Cy5, ICG, and either luciferase or luciferin.

Response: We agree with Reviewer #1's point that the size, shape, and morphology of nanoparticles are important parameters to be considered in bacterial consumption. Following Reviewer #1's helpful suggestion, high-resolution TEM images of both GP-Si-BPs and GP-Si-Luc have been added to the revised supplementary information. In addition, the TEM image and high-resolution TEM image of bare silicon nanoparticles prior to chemical conjugation of GP, Cy5, ICG, and either luciferase or luciferin have been added to the revised supplementary information (**Fig. 2a**).

Fig. 2a. TEM images and high-resolution TEM images of SiNPs, GP-Si-BPs and GP-Si-Luc. Scale bars, 20 nm and 1 nm. All imaging experiments were repeated three times with similar results.

Location of changes: Fig. 2a.

3. In Figure 1f, schematic labels are not clearly visible in the present form. It would be great to revise with letter labels rather than schematics.

Response: Following Reviewer #1's kind suggestion, we have revised the text with letter labels rather than schematics in **Fig. 2b**.

Fig. 2b. High-angle annular dark field-scanning TEM (HAADF-STEM) images of MDR *E. coli* or MRSA treated with PBS, 0.06 mM Si-BPs, Si-Luc, GP-Si-Luc, and GP-Si-BPs at 37 °C for 2.5 h. After incubation, the treated bacteria were rinsed with PBS buffer several times. The bacterial cell concentration was $\sim 1.0 \times 10^7$ CFU. Scale bar, 200 nm. All imaging experiments were repeated three times with similar results.

Location of changes: Fig. 2b.

4. Stability of luciferase and luciferin after Si NP conjugation?

Response: Accordingly, the stability of luciferase and luciferin after SiNP conjugation was systematically investigated (**Supplementary Fig. 5**). In particular, we first tested the stability of luciferase and luciferin after SiNP conjugation in a series of solutions with pH 7.5 and solutions containing various intracellular species (e.g., 150 mM KCl, 2 mM MgSO₄, 10 mM NaHCO₃, 2 mM CaCl₂, 20 mM glucose, and 1 mM bovine serum albumin (BSA)). As revealed in **Supplementary Figs. 5a, b**, the fluorescence and bioluminescence of GP-Si-BPs hardly change in these solutions.

Furthermore, we tested the stability of luciferase and luciferin after SiNP conjugation at different times and at various temperatures. Additionally, the fluorescence and bioluminescence of GP-Si-BPs hardly change at different times and at various temperatures, as revealed in **Supplementary Figs. 5c, d**. Taken together, luciferase and luciferin after SiNP conjugation exhibit good stability.

Supplementary Fig. 5. Stability test of GP-Si-BPs in nanoagents. **a**, The fluorescence intensity of ICG in GP-Si-BPs tested in various kinds of solutions with pH 7.5, as well as different kinds of solutions containing various intracellular species (e.g., 2 mM MgSO₄, 2 mM CaCl₂, 150 mM KCl, 10 mM NaHCO₃, 20 mM glucose and 1 mM bovine serum albumin (BSA)). **b**, The bioluminescence intensity of GP-Si-BPs in nanoagents is tested in various kinds of solutions with pH 7.5, as well as different kinds of solutions containing various intracellular species (e.g., 2 mM MgSO₄, 2 mM CaCl₂, 150 mM KCl, 10 mM NaHCO₃, 20 mM glucose and 1 mM bovine serum albumin (BSA)). **c**, Fluorescence intensity of GP-Si-BPs in PBS after storage at various temperatures for 24 hours and 72 hours. **d**, The bioluminescence intensity of GP-Si-BPs in PBS after storage at various temperatures for 24 hours and 72 hours. All error bars represent the standard deviation determined from three independent assays. Source data are provided as a Source Data file.

Location of changes: Supplementary Fig. 5, Paragraph 1 on Page 7.

Special thanks to Reviewer #1's comments again.

苏州大学

复旦大学附属 眼耳鼻喉科医院

EYE & ENT HOSPITAL OF FUDAN UNIVERSITY

上海市五官科医院

Address: 83 Road Fenyang, Xuhui District, Shanghai, 200031, China

Telephone: +86-021-64377134

Fax: +86-021-64377151

Website: www.fdeent.org

Reviewer #2 (Remarks to the Author):

The concept that is described in the manuscript is interesting, novel, creative and promising. To my best knowledge, this concept in bacterial imaging has never been published. The authors describe a thorough testing of this method *in vitro* and *in vivo*, which also has a large potential value.

However, in its current state the manuscript lacks several basic elements needed for transparent research, that allows for a qualitative review process. I will address these basic elements below. Subsequently, a more detail review of the first 8 pages is given. After addressing the mentioned basic elements by the authors, a further detailed review of the rest of the manuscript would be necessary.

Summary of response: We thank the reviewer for his/her positive comments regarding the novelty and significance of the manuscript and appreciate their acknowledgement of the benefits of this application in bacterial imaging. In response to the thoughtful comments by the reviewer, we have added new data and discussion to the manuscript to address his/her concerns, especially several basic elements needed for transparent research. Accordingly, the point-by-point responses to the comments made by Reviewer #2 are given below.

Below are specific comments:

1. The result section, including the figures, encompasses 20 pages. For the majority it does not only describe actual results of the current research, but also introduction, methods and hypothesis. Moreover, is not clearly marked where theoretical hypothesis transfers to actual results. This should be substantially restructured and shortened.

Response: We agree with Reviewer #2's point that the majority describe not only the actual results of the current research but also the introduction, methods and hypothesis, and it is not clearly marked where the theoretical hypothesis transfers to the actual results. Following Reviewer #2's helpful suggestion, we have substantially restructured and shortened the results section. Typically, we have removed original **Fig. 1a** from the introduction section, removed original **Fig. 1c**, **Fig. 4a**, **Fig. 4d**, **Fig. 4f**, **Fig. 4g**, **Fig. 5a**, **Fig. 5d**, **Fig. 5f**, **Fig. 5g**, **Fig. 6** and **Fig. 7a** from the supporting information, and deleted **Fig. 1b**. Additionally, the introduction, methods and hypothesis have been removed from the result section.

Location of changes: The introduction section, the result section, the discussion section, Supplementary Fig. 1, Supplementary Fig. 18, Supplementary Fig. 21, Supplementary Fig. 23 and Supplementary Fig. 24.

2. Of specific results and figures, it is unclear how these were generated. A couple of examples:

Response: Thank you for the comment. Accordingly, the details of how the specific results and figures were generated are provided in the revised manuscript. The point-by-point responses are given below:
o Lines 133-134: it is unclear how these exact numbers were derived. Was DLS used for this? In this case, suppl fig 1 should show how these numbers were generated.

Response: Accordingly, suppl fig 1 has shown how these exact numbers were generated.

Typically, the size distributions of GP-Si-BPs, GP-Si-Luc and pure SiNPs in corresponding TEM images were obtained by measuring 200 particles (**Supplementary Fig. 3**). Accordingly, the average diameter of GP-Si-BPs was ~2.8 nm, and the average diameter of GP-Si-Luc was ~2.9 nm, both of

which were slightly larger than that of naked SiNPs (e.g., ~ 2.2 nm).

The hydrodynamic diameters of GP-Si-BPs, GP-Si-Luc and pure SiNPs were ~ 5.6 nm, ~ 4.1 nm and ~ 3.1 nm, respectively, as measured by dynamic light scattering (DLS) (**Supplementary Fig. 4**). DLS was performed using a DynaPro DLS, which was made by Malvern Corp, U.K. (ZEN3690). One milliliter of GP-Si-BP, GP-Si-Luc or pure SiNP sample was transferred into an exclusive vitreous for DLS measurements. The experimental parameters were as follows: scan times: 100; dispersant: water; temperature: 25 °C; viscosity: 0.8872 cP; RI: 1.330; and dielectric constant: 78.5.

Supplementary Fig. 3. The size distribution of nanoparticles. **a**, Size distribution of SiNPs. **b**, Size distribution of GP-Si-BPs. **c**, Size distribution of GP-Si-Luc. The size distribution data were obtained by measuring 200 particles in the corresponding TEM images (**Fig. 2a**). Accordingly, the average diameter of GP-Si-BPs was ~2.8 nm, and the average diameter of GP-Si-Luc was ~ 2.9 nm, both of which were slightly larger than that of naked SiNPs (e.g., ~ 2.2 nm). Source data are provided as a source data file.

Supplementary Fig. 4. DLS analysis of SiNPs, GP-Si-Luc and GP-Si-BPs. DLS was performed using a DynaPro DLS, which was made by Malvern Corp, U.K. (ZEN3690). One milliliter of GP-Si-BP, GP-Si-Luc or pure SiNP sample was transferred into an exclusive vitreous for DLS measurements. The experimental parameters were as follows: scan times: 100; dispersant: water; temperature: 25 °C; viscosity: 0.8872 cP; RI: 1.330; and dielectric constant: 78.5. Accordingly, the hydrodynamic diameters of GP-Si-BPs, GP-Si-Luc and pure SiNPs measured by DLS were ~ 5.6 nm, ~ 4.1 nm and

苏州大学

复旦大学附属眼耳鼻喉科医院

EYE & ENT HOSPITAL OF FUDAN UNIVERSITY

上海市五官科医院

Address: 83 Road Fenyang, Xuhui District, Shanghai, 200031, China

Telephone: +86-021-64377134

Fax: +86-021-64377151

Website: www.fdeent.org

~ 3.1 nm, respectively. Source data are provided as a source data file.

Location of changes: Supplementary Fig. 3, Paragraph 2 on Page 6.

Moreover, at the small TEM images of fig 1d-e, using the scale bar, the particles seem smaller than 5.6 nm, so how this related than to the supposed DLS data is not completely clear to me and should be explained.

Response: We apologize for our misleading description. Indeed, the particles in the small TEM images using the scale bar are smaller than 5.6 nm, according to the size distribution analysis in **Supplementary Fig. 3**. Accordingly, the average diameter of GP-Si-BPs was ~2.8 nm, and the average diameter of GP-Si-Luc was ~2.9 nm, both of which were slightly larger than that of naked SiNPs (e.g., ~2.2 nm). In fact, 5.6 nm is the hydrodynamic diameter of the GP-Si-BPs, as measured by DLS. We have corrected this error in the revised manuscript. The different diameters measured by TEM and DLS are due to different surface states of the same sample under the two measurement conditions. Specifically, the solvent in the sample must be strictly removed for TEM characterization, thus yielding a smaller diameter than that measured by DLS. Similar results were also reported in other works (e.g., *J. Am. Chem. Soc.* **137**, 14726-14732 (2015); *J. Am. Chem. Soc.* **133**, 14192 (2011); etc.).

Location of changes: Paragraph 2 on Page 6.

Moreover, in Fig 1 d and e, it is not clear to me how based on this image it is concluded that the particles are indeed spherical.

Response: Accordingly, we have deleted the conclusion “the particles are indeed spherical.”

o Figure 1f shows one of the key elements of the article. However, it is not described how this figure should be interpreted.

Response: Accordingly, we have interpreted this figure clearly. Typically, as displayed in the elemental mapping in high-angle annular dark field-scanning transmission electron microscope (HAADF-STEM) images (**Fig. 2b**), carbon, nitrogen and oxygen elements appeared in each group, while silicon elements existed only in the bacteria treated with GP-Si-BPs or GP-Si-Luc. Apparently, the observed silicon signals were assigned to SiNPs in GP-Si-BPs or GP-Si-Luc, thus directly demonstrating the internalization of GP-Si-BPs or GP-Si-Luc into bacterial cells.

Location of changes: Paragraph 2 on Page 7.

o Suppl fig 3a: a linear correlation cannot be claimed by the number and spacing of the measured points.

Response: We agree. Accordingly, a new good linear correlation ($R^2=0.997$) is shown in the new **Supplementary Fig. 7a**.

Supplementary Fig. 7. a, Linear correlation between luminescence intensities and ATP concentrations in the ATP kit assay.

Location of changes: Supplementary Fig. 7a.

o Suppl fig 3b: it is not clear how the data for this figure was generated. Methods are lacking.

Response: Accordingly, the methods for the generation of the data in this figure have been added in the revised manuscript. Typically, after centrifuging the bacterial solution, we used the ATP detection kit to measure the bioluminescence intensity of the supernatant and the bacterial solution after equal volume PBS reselection and then inserted the detected bioluminescence intensity into the linear regression equation in **Supplementary Fig. 7a** to calculate the ATP content.

Location of changes: Legend in Supplementary Fig. 7b.

3. Specific methods, such as ATP assay and DCC, are not included in the methods. Please include all methods.

Response: Following Reviewer #2's helpful suggestion, we added the methods of ATP assay and DCC in the methods section. Typically, the luminescence signals of GP-Si-BPs in different solutions were examined using the IVIS Spectrum imaging system. To this end, 0.06 mM GP-Si-BPs were mixed with different concentrations of ATP in a black 96-well plate. Immediately after thorough mixing, the plate was placed into the imaging system to obtain the luminescence signal. Similarly, the decayed luminescence of GP-Si-BPs was quantified after incubation with the ATP inhibitor DCC in the presence of GP-Si-Luc.

Location of changes: Paragraph 2 on Page 28

4. The figures are cluttered and not well structured. It is difficult to get the intended message from the figures. In footnotes of figures, it should be explained what can be seen in the figure. For example, in fig 1f and suppl fig 2, this is not clear. On the other side, repetition of the methods is not specifically necessary in footnotes.

Response: We agree that the figures are cluttered and not well structured. Following Reviewer #2's helpful suggestion, we have revised the footnotes of **Fig. 2b** and **Supplementary Fig. 6**. The revised footnotes could explain what can be seen in the figure. The repetition of the methods is not included in footnotes.

Fig. 2b. High-angle annular dark field-scanning TEM (HAADF-STEM) images of MDR *E. coli* or MRSA treated with PBS, 0.06 mM Si-BPs, Si-Luc, GP-Si-Luc, and GP-Si-BPs at 37 °C for 2.5 h. After incubation, the treated bacteria were rinsed with PBS buffer several times. The bacterial cell concentration was $\sim 1.0 \times 10^7$ CFU. Scale bar, 200 nm. All imaging experiments were repeated three times with similar results.

Supplementary Fig. 6. SEM images of bacteria treated with Trojan BLI probes. **a**, SEM images of MRSA or MDR *E. coli* treated with PBS, 0.06 mM Si-BPs or GP-Si-BPs at 37 °C for 2.5 h. After incubation, the treated bacteria were rinsed with PBS buffer several times. **b**, SEM images of MRSA or MDR *E. coli* treated with PBS, 0.06 mM Si-Luc or GP-Si-Luc at 37 °C for 2.5 h. After incubation, the treated bacteria were rinsed with PBS buffer several times. The bacterial cell concentration was $\sim 1.0 \times 10^7$ CFU. Scale bar, 200 nm.

Location of changes: Fig. 2b, Supplementary Fig. 6.

5. The research is of additive value if the researcher intends to use the probe in specific clinical settings *in vivo*. In the results and discussion this is suggested, e.g., by testing a probe on human clinical samples (in this case vitreous), and when referring to FDA approval, respectively. This should be more thoroughly addressed and discussed, including limitations.

Response: Thanks a lot for Reviewer #2's significant suggestions. Accordingly, we have thoroughly addressed and discussed the potential use of the strategy in specific clinical settings *in vivo*, including limitations. Typically, in our design, the bioluminescent indicators were loaded onto GP-modified SiNPs and transported into bacteria through ABC sugar transporters. Of note, SiNPs, as vectors, have been extensively employed in various bioapplications due to their low/no toxicity (e.g., *Adv. Mater.* **27**, 1029-1034 (2015); *Nano Today*, **26**, 149-163 (2019); *Adv. Mater.* **28**, 10567-10574 (2016); *Acc. Chem. Res.* **47**, 612-623 (2014)). As previously reported, SiNPs could be biodegraded and renally cleared

苏州大学

复旦大学附属眼耳鼻喉科医院

EYE & ENT HOSPITAL OF FUDAN UNIVERSITY

上海市五官科医院

Address: 83 Road Fenyang, Xuhui District, Shanghai, 200031, China

Telephone: +86-021-64377134

Fax: +86-021-64377151

Website: www.fdeent.org

without distinct toxicity *in vivo* (e.g., *Nat. Mater.* **8**, 331–336 (2009); *Nat. Commun.* **4**, 2326 (2013)). For instance, toxicity assessments in *Caenorhabditis elegans*, silkworm and even nonhuman primate animal models (e.g., cynomolgus macaques) have demonstrated that internalized SiNPs do not affect the morphology, physiology or innate immune functions of animals, suggesting the benign biocompatibility of SiNPs in live organisms (e.g., *Chemosphere* **159**, 628-637 (2016); *Nano Res.* **11**, 2336-2346 (2017); *Nano Res.* **14**, 3840–3847 (2021)). On this basis, small SiNPs received FDA-approved investigational new drug approval for the first-in-human clinical trial in January 2011 (i.e., NCT01266096, NCT02106598) (e.g., *Sci. Trans. Med.* **6**, 260ra149 (2014); *J. Clin. Invest.* **121**, 2768-2780 (2011)). As such, we selected SiNPs as vehicles to deliver bioluminescent indicators. The bioluminescent indicators included luciferase and two organic dyes, Cy5 and ICG. As a result, NIR emission can be achieved using a dual resonance energy transfer relay process that combines BRET between luciferase and Cy5 and FRET between Cy5 and ICG. Therefore, the developed strategy has the potential to image bacteria in deep tissues in specific clinical settings. Notwithstanding, the combination of luciferase, Cy5, and ICG requires relatively complicated work for sample preparation and quantitative analysis, which significantly lowers the impact of this work. In the following study, we will test a simpler system.

Location of changes: Paragraph 3 on Page 25, Paragraph 1 on Page 26.

6. In extension of a supposed clinical use of probe, stability testing is lacking.

Response: Accordingly, the stability of luciferase and luciferin after SiNP conjugation was systematically investigated (**Supplementary Fig. 5**). In particular, we first tested the stability of luciferase and luciferin after SiNP conjugation in a series of solutions with pH 7.5 and solutions containing various intracellular species (e.g., 150 mM KCl, 2 mM MgSO₄, 10 mM NaHCO₃, 2 mM CaCl₂, 20 mM glucose, and 1 mM bovine serum albumin (BSA)). As revealed in **Supplementary Figs. 5a, b**, the fluorescence and bioluminescence of GP-Si-BPs hardly change in these solutions. Furthermore, we tested the stability of luciferase and luciferin after SiNP conjugation at different times and at various temperatures. Additionally, the fluorescence as well as the bioluminescence of GP-Si-BPs hardly change at different times and at various temperatures, as revealed in **Supplementary Figs. 5c, d**. Taken together, luciferase and luciferin after SiNP conjugation exhibit good stability.

Supplementary Fig. 5. Stability test of GP-Si-BPs in nanoagents. **a**, The fluorescence intensity of GP-Si-BPs in nanoagents is tested in various kinds of solutions with pH 7.5, as well as different kinds of solutions containing various intracellular species (e.g., 2 mM MgSO₄, 2 mM CaCl₂, 150 mM KCl, 10 mM NaHCO₃, 20 mM glucose and 1 mM bovine serum albumin (BSA)). **b**, The bioluminescence intensity of GP-Si-BPs in nanoagents is tested in various kinds of solutions with pH 7.5, as well as different kinds of solutions containing various intracellular species (e.g., 2 mM MgSO₄, 2 mM CaCl₂, 150 mM KCl, 10 mM NaHCO₃, 20 mM glucose and 1 mM bovine serum albumin (BSA)). **c**, Fluorescence intensity of GP-Si-BPs in PBS after storage at various temperatures for 24 hours and 72 hours. **d**, The bioluminescence intensity of GP-Si-BPs in PBS after storage at various temperatures for 24 hours and 72 hours. All error bars represent the standard deviation determined from three independent assays. Source data are provided as a Source Data file.

Location of changes: Supplementary Fig. 5, Paragraph 1 on Page 7.

7. The introduction and discussion should include a thorough description and consideration of the BRET/FRET concept.

Response: Following your helpful suggestion, the introduction and discussion have included a thorough description and consideration of the BRET/FRET concept.

The introduction section: By further employing an energy transfer relay that integrates bioluminescence resonance energy transfer (BRET) between luciferase and Cy5 and fluorescence resonance energy transfer (FRET) between Cy5 and ICG, the developed Trojan BLI probes enabled near-infrared (NIR) imaging of bacteria within deep tissues.

苏州大学

复旦大学附属眼耳鼻喉科医院

EYE & ENT HOSPITAL OF FUDAN UNIVERSITY

上海市五官科医院

Address: 83 Road Fenyang, Xuhui District, Shanghai, 200031, China

Telephone: +86-021-64377134

Fax: +86-021-64377151

Website: www.fdeent.org

The discussion section: The bioluminescent indicators included luciferase and two organic dyes, Cy5 and ICG. As a result, NIR emission can be achieved using a dual resonance energy transfer relay process that combines BRET between luciferase and Cy5 and FRET between Cy5 and ICG. Therefore, the developed strategy has the potential to image bacteria in deep tissues in specific clinical settings. Notwithstanding, the combination of luciferase, Cy5, and ICG requires relatively complicated work for sample preparation and quantitative analysis, which significantly lowers the impact of this work. In the following study, we will test a simpler system.

Location of changes: Paragraph 2 on Page 4, Paragraph 1 on Page 5, Paragraph 2 on Page 26.

8. Check English grammar.

Response: Thank you for the reviewer's suggestion. We have checked the full text and modified the sentences with grammar problems. Additionally, we used the 'AJE's AI Editing' service released by Nature to polish the manuscript.

9. Whether the research described in this manuscript is novel enough compared to earlier published work by the same authors in *Nat com*, is questionable. However, the described concept in itself is indeed very novel.

Response: Accordingly, the advance and novelty of the paper has been further investigated based on a deeper dive into the literature. To simplify the discussion, please see below a table summarizing the features of our previous works based on glucose polymer (GP)-modified nanoagents. As summarized in **Table S1**, one key concept in these works is that GP-modified nanoagents can be readily internalized into various bacteria through the ABC transporter pathway. Typically, in the first paper (*Nat. Commun.* **10**, 4057 (2019)), we used nanoagents made of GP- and Ce6-modified SiNPs for fluorescence imaging and photodynamic killing of various bacteria in skin tissues for the first time. On the basis of this work, we developed the first example of photoacoustic imaging and photodynamic and photothermal killing of various bacteria in gut and tumour tissues by using GP- and Ce6-modified gold nanoparticles (AuNPs) (e.g., *Nat. Commun.* **13**, 1255 (2022)). In addition to bacterial detection and inactivation, we used GP-modified nanoagents to construct the first bacteria-based drug delivery system for cancer treatment (e.g., *Nat. Commun.* **13**, 5127 (2022); *Angew. Chem. Int. Ed.* **61**, e202208422 (2022)). Apparently, these works sufficiently show technical advancement and strategic novelty and have been published in *Nature Communications* and related high-standard journals. Likewise, in this work, we present **the first demonstration of bioluminescence imaging and photothermal killing of various bacteria in deep tissues by using GP-modified nanoprob**. Specifically, the developed nanoprobe is totally different from what we have reported before. Different from other BLI methods for bacterial detection (e.g., *Sci. Adv.* **7**, eaaz9857 (2021); *Nat. Protoc.* **3**, 629–636 (2008); *Nat. Biotechnol.* **22**, 313–320 (2004); *etc.*), **this is the first paper to use BLI to visualize various natural bacteria *in vivo*, based on the delivery of bioluminescence reports to bacterial cells that directly consume ATP inside the bacteria.** Taken together, we believe the presented work fully shows technical advancement and strategic novelty to be fitted for a high standard profile in *Nature Communications*.

Table S1. Comparison of the features of glucose polymer (GP)-modified agents.

Refs.	Features		
	Agents composition	Mechanisms of entry into bacteria	Functions
Nat. Commun. 10 , 4057 (2019)	GP and chlorin e6 (Ce6) modified silicon nanoparticles (SiNPs)	ABC transporter pathway	Fluorescence imaging and photodynamic killing of gram-negative and gram-positive bacteria
Nat. Commun. 13 , 1255 (2022)	GP and Ce6 modified gold nanoparticles (AuNPs)	ABC transporter pathway	Photoacoustic imaging and photodynamic and photothermal killing of gram-negative and gram-positive bacteria
Nat. Commun. 13 , 5127 (2022)	GP and indocyanine green (ICG) modified SiNPs	ABC transporter pathway	A bacteria-based drug delivery system for glioblastoma photothermal immunotherapy
Angew. Chem. Int. Ed. 61 , e202208422 (2022)	GP and ICG modified SiNPs, AuNPs or carbon dots (CDs)	ABC transporter pathway	A bacteria-based drug delivery system for photothermal programmable destruction of deep tumor tissues
This work	GP, luciferase, Cy5 and ICG-modified SiNPs, GP and D-luciferin-modified SiNPs	ABC transporter pathway	Bioluminescence imaging and photothermal killing of gram-negative and gram-positive bacteria

Location of changes: Supplementary Table 1, Paragraph 2 on Page 4.

10. The title claims “Deep organ” imaging. This is not described in the manuscript and could be considered misleading.

Response: Thank you very much for the suggestion. We have modified “deep organ” to “deep tissue”.

Location of changes: The title.

11. The advantage of BLI imaging without genetical engineering and BLI imaging without a need of damaging the bacterial cells, by using a Trojan horse technique could be clearer addressed in the last sentence of the abstract.

苏州大学

复旦大学附属眼耳鼻喉科医院

EYE & ENT HOSPITAL OF FUDAN UNIVERSITY

上海市五官科医院

Address: 83 Road Fenyang, Xuhui District, Shanghai, 200031, China

Telephone: +86-021-64377134

Fax: +86-021-64377151

Website: www.fdeent.org

Response: Following Reviewer 2's helpful suggestion, the last sentence of the abstract has been addressed in a clearer manner: "The proposed strategy delivers luciferase and luciferin into various bacteria for *in vivo* visualization without any genetic modification of bacteria and without destroying the bacterial cells".

Location of changes: Abstract.

12. The probe is tested in clinical materials. The term "human testing" is misleading.

Response: Following the reviewer's significant suggestion, we have avoided using the term "human testing".

13. Chemical yields are lacking in synthesis.

Response: Accordingly, the chemical yield of GP-Si-BPs was 76.1%, and the chemical yield of GP-Si-Luc was 78.8% according to the reported protocol (e.g., *Adv. Mater.* **27**, 1029–1034 (2015)).

Location of changes: Paragraph 1 on Page 6.

14. Line 29: add "in vitro".

Response: Following the reviewer's suggestion, we have added "*in vitro*" in line 29.

Location of changes: Paragraph 1 on Page 2.

15. Line 46-48: "A key ... host organism" the meaning of this sentence was not clear for me.

Response: Following the reviewer's suggestion, we have modified the sentence "A key ... host organism" as "Bacterial activity *in vivo* is heavily influenced by their location within the host organism".

Location of changes: Paragraph 1 on Page 3.

16. Line 49 and 60: please avoid the "and so forth".

Response: Following the reviewer's suggestion, we have deleted "and so forth".

Location of changes: Paragraph 2 on Page 3

17. Line 50-52: infection is not an inflammation symptom, which is implied by the formulation of this sentence.

Response: Accordingly, we have modified the corresponding sentence: "However, due to their relatively poor selectivity, they are unable to distinguish inflammation caused by bacterial infections from inflammation caused by other causes such as cancer or autoimmune diseases."

Location of changes: Paragraph 1 on Page 3.

18. Line 54-57: near infrared imaging has not such a large autofluorescence problem as described in this sentence. The certainty of the sentence should be tuned down/nuanced.

Response: Accordingly, the certainty of the sentence has been tuned down: "As the most widely used optical imaging method, fluorescence imaging requires real-time optical excitation; however, it **may** lead to background autofluorescence of biological tissues, resulting in a **relatively** poor signal-background ratio."

Location of changes: Paragraph 1 on Page 3.

19. Line 68-70: first, the...pathogenic bacteria” This is a non-sensical sentence. It was already stated that non-genetically engineered bacteria lack BLI. Therefore, it is logical that BLI is naturally not present in commensal and pathogenic bacteria. Furthermore, if BLI is not present, it is not “difficult to visualize endogenous BLI systems” when not present, but that is intrinsically impossible.

Response: We agree. We have revised this nonsensical sentence as follows: “first, the endogenous BLI systems need genetic modification of bacteria”.

Location of changes: Paragraph 2 on Page 3.

20. Line 70-72: This is a non-sensical sentence. Please rephrase where it is more clearly addressed that the availability of extracellular ATP is a major limitation for imaging live bacteria.

Response: Accordingly, we have revised the nonsensical sentence as follows: “Second, exogenous BLI systems need the destruction of bacterial cells to consume intracellular ATP, thus being unable to image live bacteria.”

Location of changes: Paragraph 2 on Page 3.

21. Line 82-83: please explain and/or reference why α (1-4)-glucosidically linked glucose polymers (dextrose equivalent of 4.0~7.0)-linked nanoparticles are used.

Response: Following the reviewer’s suggestion, the relevant explanation has been added to the revised manuscript. Routinely, GP (e.g., *poly[4-O-(α -D-glucoopyranosyl)-D-glucoopyranose]*) serves as the ubiquitous carbon source and can be robustly internalized into bacterial cells through the ABC sugar transporter (e.g., *Nat. Commun.* **11**, 1250 (2020); *Nat. Commun.* **10**, 4057 (2019); *Nat. Commun.* **13**, 1255 (2022)). For example, the ABC sugar transporter in *Escherichia coli* (*E. coli*) has five subunits: LamB, MalE, MalF, MalG and MalK. Among these subunits, LamB is a typical outer membrane diffusion porin, and MalE can recognize α (1-4)-glucosidically linked GP molecules (e.g., *Angew. Chem. Int. Ed.* **61**, e202208422 (2022); *Nat. Commun.* **13**, 5127 (2022); *Angew. Chem. Int. Ed.* **53**, 14096-14101 (2014); *Mol. Microbiol.* **77**, 1354-1366 (2010); *Ann. Microbiol.* **133A**, 153-159 (1982); *Res. Microbiol.* **153**, 417-424 (2002); *Mol. Biol. Rev.* **62**, 204-229 (1998)). By leveraging this uptake mechanism, small-size (e.g., ~5 nm) GP-modified nanoparticles, including silicon nanoparticles, gold nanoparticles and carbon dots, have recently been demonstrated to be selectively and robustly internalized into bacterial cells (e.g., *J. Biol. Chem.* **263**, 314-320 (1988); *PLoS One* **5**, e10349 (2010); *J. Biol. Chem.* **268**, 18617-18621 (1993)). Analogously, diverse bacteria eat their camouflaged ‘foods’, i.e., GP-Si-BPs and GP-Si-Luc.

Location of changes: Paragraph 2 on Page 4.

22. Line 84: please replace “kinds” by “species” or “strains”.

Response: Following the reviewer’s suggestion, we have replaced “kinds” with “species” or “strains”.

Location of changes: Paragraph 1 on Page 2, Paragraph 1 on Page 5

23. Line 138: “human-derived” could be rephrased, e.g., as clinically derived.

Response: Following the reviewer’s suggestion, we have rephrased “human-derived” as “clinically

derived”.

Location of changes: Paragraph 2 on Page 7, Paragraph 3 on Page 12, Paragraph 2 on Page 26, Paragraph 1 on Page 29.

24. Line 138: Abbreviations, such as MDR and MRSA, should be written out in full.

Response: Following the reviewer’s suggestion, the abbreviations, such as MDR and MRSA, have been written out in full.

Location of changes: Paragraph 2 on Page 7.

25. Line 145: surface and morphology are not “identical”, but could be described as comparable.

Response: Following the reviewer’s suggestion, “identical” has been revised to “comparable”.

Location of changes: Paragraph 2 on Page 7.

26. Line 146: “election” should be “electron”?

Response: Following the reviewer’s suggestion, “election” has been revised to “electron”.

Location of changes: Paragraph 2 on Page 7.

27. Figure 1a: this illustrates the hypothesis, but is not a result. It should be uncoupled from the results section.

Response: We agree. **Fig. 1** has been uncoupled from the results section.

Fig. 1. Schematic design of ABC sugar transporter enabling selective delivery of bioluminescent

nanoprobes into gram-positive bacteria and gram-negative bacteria to visualize various natural bacteria *in vivo* with bioluminescence by directly consuming the ATP inside the bacteria. The nanoprobes are made of GP, Cy5, ICG and luciferase-modified silicon nanoparticles (SiNPs) (GP-Si-BPs) and GP, D-luciferin-modified SiNPs (GP-Si-Luc).

Location of changes: Fig. 1.

28. Figure 1b: the added value of this figure is unclear.

Response: We agree. Accordingly, we removed this figure in the revised manuscript.

29. Figure 1f shows one of the key elements of the article. However, it is not described how this figure should be interpreted.

Response: Accordingly, we have interpreted this figure clearly. Typically, as displayed in the elemental mapping in high-angle annular dark field-scanning transmission electron microscope (HAADF-STEM) images (**Fig. 2b**), carbon, nitrogen and oxygen appeared in each group, while silicon existed only in the bacteria treated with GP-Si-BPs or GP-Si-Luc. Apparently, the observed silicon signals were assigned to SiNPs in GP-Si-BPs or GP-Si-Luc, thus directly demonstrating the internalization of GP-Si-BPs or GP-Si-Luc into bacterial cells.

Location of changes: Paragraph 2 on Page 7.

30. Figure 1g: the control including GP-Si-Luc with Si-BP is lacking. Please add this. Moreover, the lysed positive control with GP-Si-BPs and GP-Si-Luc might be of additive value and should be added.

Response: Following the reviewer's helpful suggestion, the control including GP-Si-Luc with Si-BP and the lysed positive control with GP-Si-BPs and GP-Si-Luc has been added to **Fig. 2c**.

Fig. 2c. Bioluminescence signals of MDR *E. coli* with different treatments as indicated. After incubation, the treated bacteria were rinsed with PBS buffer several times, followed by imaging (IVIS Lumina III).

Location of changes: Fig. 2c.

苏州大学

复旦大学附属眼耳鼻喉科医院

EYE & ENT HOSPITAL OF FUDAN UNIVERSITY

上海市五官科医院

Address: 83 Road Fenyang, Xuhui District, Shanghai, 200031, China

Telephone: +86-021-64377134

Fax: +86-021-64377151

Website: www.fdeent.org

Special thanks to Reviewer #2's comments again.

Finally, we thank you very much for the editor's and reviewers' valuable comments, which vastly facilitated improvement of the quality of this manuscript, making it possible to satisfy the requirements of the esteemed journal Nature Communications. Thank you very much!

REVIEWERS' COMMENTS

Reviewer #1 (Remarks to the Author):

Authors revised the manuscript to improve the overall research output accordingly to the reviewers' comments. It is very impressive that authors have shown their great endeavour to revise the manuscript in a short period of time. I'm pleased to review this work which includes substantial improvement to clarify the importance of research with supportive schematics, tables, and plots. All the work seems to be supportive with data, and I believe that the authors provide their best effort on this manuscript to be published in Nature Communications.

One last concern is still about the novelty of this work compared to their previous studies. Although the authors provided point-by-point revision, particularly with a supplementary table, major concept of this study is still in a series of their work on GP-modified agents for bacterial uptake and imaging. Despite of the systematic investigation given by the authors, the reviewer's suggestion on this manuscript is still against to publication in Nature Communications because of the novelty and significance. However, this could be finally evaluated by the editor.

Reviewer #2 (Remarks to the Author):

Methodology is still not consistently described in the main manuscript.

Comments were specifically addressed to the mentioned issues but not extrapolated and similar issues still exist throughout the manuscript

Reviewer #3 (Remarks to the Author):

I am reading a revised manuscript, which aims to tackle the difficulty in imaging natural bacteria using bioluminescence imaging technology. The authors take advantage of bacteria-specific ATP-binding cassette (ABC) sugar transporters to enable selective delivery of bioluminescent nanoprobe into gram-positive bacteria and gram-negative bacteria to visualize various natural bacteria in vivo. The developed nanoprobe even allows to differentiate bacterial and nonbacterial nephritis and colitis in mice. This is indeed a creative work to the community of in vivo imaging. The authors also have made their

efforts to answer the questions from the previous reviewers and the quality has been improved. Thus, I recommend its publication after addressing the following minor issues.

1. The main contribution of this work is the development of a nanoprobe. However, there is no relevant description related to the chemistry and design of the nanoprobe. Thus, I feel the abstract needs to be tailored a bit to orient to the imaging community as well as to the materials community.

2. This also needs to be taken care of the introduction. The authors may cite the previous literatures on developing self-luminescence imaging probes for in vivo detection. For instance, *Chemical Reviews*, 2021, 121, 13086-1313; *Nature Biotechnology*, 2017, 35, 1102; *Nature Biomedical Engineering*, 2023, 7, 10.1038/s41551-023-01009-1.

Address: 83 Road Fenyang, Xuhui District, Shanghai, 200031, China
Fax: +86-021-64377151

复旦大学附属眼耳鼻喉科医院
EYE & ENT HOSPITAL OF FUDAN UNIVERSITY
上海市五官科医院

Telephone: +86-021-64377134
Website: www.fdeent.org

Response to Reviewers' comments

Reviewer #1 (Remarks to the Author):

Authors revised the manuscript to improve the overall research output accordingly to the reviewers' comments. It is very impressive that authors have shown their great endeavour to revise the manuscript in a short period of time. I'm pleased to review this work which includes substantial improvement to clarify the importance of research with supportive schematics, tables, and plots. All the work seems to be supportive with data, and I believe that the authors provide their best effort on this manuscript to be published in Nature Communications.

One last concern is still about the novelty of this work compared to their previous studies. Although the authors provided point-by-point revision, particularly with a supplementary table, major concept of this study is still in a series of their work on GP-modified agents for bacterial uptake and imaging. Despite of the systematic investigation given by the authors, the reviewer's suggestion on this manuscript is still against to publication in Nature Communications because of the novelty and significance. However, this could be finally evaluated by the editor.

Response: We thank Reviewer#1's positive comments. We believe the revised manuscript fully shows the novelty and significance to be fitted for a high standard profile in Nature Communications. And we also agree with Reviewer#1's comment that this could be finally evaluated by the editor.

Reviewer #2 (Remarks to the Author):

Methodology is still not consistently described in the main manuscript.

Comments were specifically addressed to the mentioned issues but not extrapolated and similar issues still exist throughout the manuscript

Response: We thank Reviewer#2's comments. Accordingly, the manuscript has been carefully checked and edited to ensure consistent descriptions of methodology throughout.

Reviewer #3 (Remarks to the Author):

I am reading a revised manuscript, which aims to tackle the difficulty in imaging natural bacteria using bioluminescence imaging technology. The authors take advantage of bacteria-specific ATP-binding cassette (ABC) sugar transporters to enable enabling selective delivery of bioluminescent nanoprobe into gram-positive bacteria and gram-negative bacteria to visualize various natural bacteria in vivo. The developed nanoprobe even allows to differentiate bacterial and nonbacterial nephritis and colitis in mice. This is indeed a creative work to the community of in vivo imaging. The authors also have made their efforts to answer the questions from the previous reviewers and the quality has been improved. Thus, I recommend its publication after addressing the following minor issues.

Summary of response: We thank the reviewer for his/her positive comments regarding the novelty and significance of the manuscript and appreciate their acknowledgement of the benefits of this application in bacterial imaging. Accordingly, the point-by-point responses to the comments made by Reviewer #3 are given below.

1. The main contribution of this work is the development of a nanoprobe. However, there is no relevant description related to the chemistry and design of the nanoprobe. Thus, I feel the abstract needs to be tailored a bit to orient to the imaging community as well as to the materials community.

复旦大学附属 眼耳鼻喉科医院
EYE & ENT HOSPITAL OF FUDAN UNIVERSITY
上海市五官科医院

Address: 83 Road Fenyang, Xuhui District, Shanghai, 200031, China

Telephone: +86-021-64377134

Fax: +86-021-64377151

Website: www.fdeent.org

Response: Following the reviewer's helpful suggestion, the abstract has been tailored a bit to add the description related to the chemistry and design of the nanoprobe, orienting to the imaging community as well as to the materials community: *"Typically, the synthesized bioluminescent probes are made of glucose polymer (GP), luciferase, Cy5 and ICG-modified silicon nanoparticles and their substrates are made of GP and D-luciferin-modified silicon nanoparticles."*

Location of changes: Abstract.

2. This also needs to be taken care of the introduction. The authors may cite the previous literatures on developing self-luminescence imaging probes for *in vivo* detection. For instance, *Chemical Reviews*, 2021, 121, 13086-1313; *Nature Biotechnology*, 2017, 35, 1102; *Nature Biomedical Engineering*, 2023, 7, 10.1038/s41551-023-01009-1.

Response: Following the reviewer's valuable suggestion, we have cited the previous literatures on developing self-luminescence imaging probes for *in vivo* detection.

Location of changes: Refs. 26-28.

Finally, we thank you very much for the editor's and reviewers' valuable comments, which vastly facilitate improvement of the quality of this manuscript, making it possible to satisfy requirement of the esteemed journal--- Nature Communications. Thank you very much!